# B cell-derived GABA elicits IL-10⁺ macrophages to limit anti-tumour immunity

Baihao Zhang[1], Alexis Vogelzang[1,11], Michio Miyajima[1,11], Yuki Sugiura[2,11], Yibo Wu[3], Kenji Chamoto[4], Rei Nakano[5], Ryusuke Hatae[4], Rosemary J. Menzies[4], Kazuhiro Sonomura[4], Nozomi Hojo[6], Taisaku Ogawa[6], Wakana Kobayashi[1], Yumi Tsutsui[1], Sachiko Yamamoto[1], Mikako Maruya[1], Seiko Narushima[1], Keiichiro Suzuki[1], Hiroshi Sugiya[5], Kosaku Murakami[7], Motomu Hashimoto[7], Hideki Ueno[8], Takashi Kobayashi[9], Katsuhiro Ito[4,9], Tomoko Hirano[4], Katsuyuki Shiroguchi[6], Fumihiko Matsuda[4], Makoto Suematsu[2], Tasuku Honjo[4] & Sidonia Fagarasan[1,10 ✉]

Small, soluble metabolites not only are essential intermediates in intracellular biochemical processes, but can also influence neighbouring cells when released into the extracellular milieu[1–3]. Here we identify the metabolite and neurotransmitter GABA as a candidate signalling molecule synthesized and secreted by activated B cells and plasma cells. We show that B cell-derived GABA promotes monocyte differentiation into anti-inflammatory macrophages that secrete interleukin-10 and inhibit CD8⁺ T cell killer function. In mice, B cell deficiency or B cell-specific inactivation of the GABA-generating enzyme GAD67 enhances anti-tumour responses. Our study reveals that, in addition to cytokines and membrane proteins, small metabolites derived from B-lineage cells have immunoregulatory functions, which may be pharmaceutical targets allowing fine-tuning of immune responses.

Lymphocytes are regulated v a variety of receptor interactions with soluble and cell-bound proteins. However, small metabolites derived from immune cells are also abundant in certain tissues, and many may have signalling potential that has yet to be understood. A growing body of research addresses the flux in metabolic products produced and consumed by different immune cells in various stages of differentiation and activation[1–3]. We hypothesized that water-soluble metabolites can serve as environmental cues, and mediate interactions between immune cells.

## GABA is a B cell-associated metabolite

Contrasting homeostatic (non-draining, contralateral; cLN) and activated (draining, ipsilateral; iLN) lymph nodes (LNs) were generated from mice using classic foot-pad immunization with ovalbumin (OVA) protein emulsified in complete Freud's adjuvant (CFA), and subjected to non-targeted profiling of water-soluble metabolites (Fig. 1a). Principal-component analysis revealed that a strong metabolic shift separated iLNs from cLNs in wild-type (WT) mice (Fig. 1b). Pathway analysis of around 200 metabolites with significantly different abundance between iLNs and cLNs revealed that the alanine, aspartate and glutamate pathway was the strongest metabolic feature differentiating resting and activated immune sites (Fig. 1c). Purine

and pyrimidine metabolism and the tricarboxylic acid (TCA) cycle were also strongly associated with immune activation (Fig. 1c and Extended Data Fig. 1a).

We assessed the contribution of the main lymphocyte lineages to the metabolic landscape by performing metabolome analyses on activated and resting LNs from immunodeficient mice lacking T cells ($Cd3e^{-/-}$), B cells ($Ighm^{-/-}$; referred to hereafter as $muMt^{-/-}$) or all mature T and B cells ($Rag1^{-/-}$). Despite the presence of many pathogen-associated molecules in CFA expected to stimulate pattern recognition receptors on myeloid cell subsets, the iLN profile of $Rag1^{-/-}$ mice was similar to that of cLNs, suggesting that lymphocyte activation is the dominant factor contributing to the metabolic shift in this acute inflammatory model (Fig. 1b and Extended Data Fig. 1a). B cells strongly influenced the immunized LN metabolic landscape, as $muMt^{-/-}$ samples were distinct from their WT and $Cd3e^{-/-}$ counterparts (Fig. 1b). The neurotransmitter GABA (γ-aminobutyric acid), not previously known to be synthesized by B cells, was identified as the major metabolite upregulated in iLNs in a B cell-dependent manner with respect to both fold change and $P$ value (Fig. 1d). GABA was also detected in resting cLNs from WT and $Cd3e^{-/-}$ mice, at lower levels (Fig. 1e). However, very little GABA could be detected in LNs from either B cell-deficient mice ($muMt^{-/-}$) or $Rag1^{-/-}$ mice, indicating that GABA is a signature B cell metabolite, which was confirmed in random forest algorithm analyses (Fig. 1e and

[1]Laboratory for Mucosal Immunity, Center for Integrative Medical Sciences, RIKEN Yokohama Institute, Yokohama, Japan. [2]Department of Biochemistry and Integrative Biology, Keio University, Tokyo, Japan. [3]YCI Laboratory for Next-Generation Proteomics, Center for Integrative Medical Sciences, RIKEN Yokohama Institute, Yokohama, Japan. [4]Center for Genomic Medicine, Kyoto University Graduate School of Medicine, Kyoto University, Kyoto, Japan. [5]Laboratory of Veterinary Biochemistry, Department of Veterinary Medicine, Nihon University College of Bioresource Sciences, Fujisawa, Japan. [6]Laboratory for Prediction of Cell Systems Dynamics, RIKEN Center for Biosystems Dynamics Research (BDR), Osaka, Japan. [7]Department of Rheumatology and Clinical Immunology, Kyoto University Graduate School of Medicine, Kyoto University, Kyoto, Japan. [8]Department of Immunology, Kyoto University Graduate School of Medicine, Kyoto University, Kyoto, Japan. [9]Department of Urology, Kyoto University Graduate School of Medicine, Kyoto University, Kyoto, Japan. [10]Division of Integrated High-Order Regulatory Systems, Center for Cancer Immunotherapy and Immunobiology, Kyoto University Graduate School of Medicine, Kyoto University, Kyoto, Japan. [11]These authors contributed equally: Alexis Vogelzang, Michio Miyajima, Yuki Sugiura. ✉e-mail: sidonia.fagarasan@riken.jp

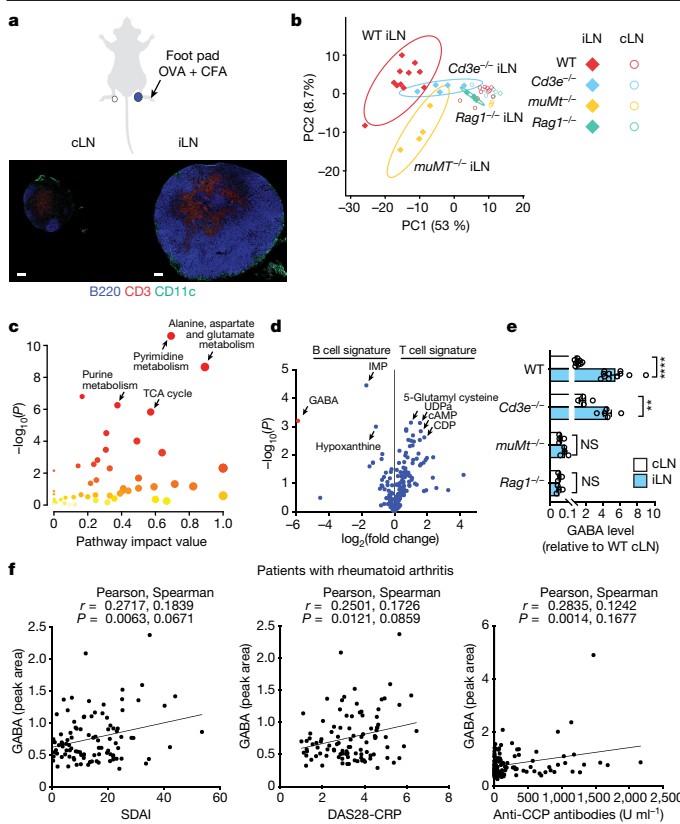

**Fig. 1 | Metabolic remodelling of immunized LNs and B cell-dependent GABA production. a–e**, Mice were injected in the foot pad with OVA + CFA, and iLNs and cLNs were collected for metabolite MS and histology at day 7: WT (*n* = 11), *Cd3*⁻/⁻ (*n* = 5), *muMt*⁻/⁻ (*n* = 4) and *Rag1*⁻/⁻ (*n* = 3). **a**, Immunohistochemistry of B cells (B220), T cells (CD3) and myeloid cells (CD11c). Scale bars, 200 μm. **b**, Principal-component analysis of metabolites in iLNs and cLNs. PC, principal component. **c**, Pathway analysis of metabolites with significantly different abundance between iLNs and cLNs in WT mice (two-tailed unpaired *t*-test, *P* < 0.05). **d**, Relative *Cd3*⁻/⁻ (B cell signature) and *muMt*⁻/⁻ (T cell signature) iLN metabolites, showing only metabolites that also differed between iLNs and cLNs in WT mice. **e**, GABA levels in the indicated cLNs or iLNs, relative to WT cLNs (data are shown as mean ± s.e.m.; two-tailed unpaired *t*-test: ***P* < 0.001, ****P* < 0.0001; NS, not significant). **f**, Correlation of plasma GABA levels with disease activity scores (Simplified Disease Activity Index (SDAI) or Activity Score 28 using C-reactive protein (DAS28-CRP)) and plasma anti-cyclic citrullinated peptide (CCP) antibody levels in patients with rheumatoid arthritis (*n* = 138). Pearson's and Spearman's *r* and *P* values (two tailed) are shown. *n* indicates the number of biological replicates. Data are representative of two experiments (**a**–**e**). Exact *P* values are provided in the Source Data.

Extended Data Fig. 1c). Imaging mass spectrometry (IMS) confirmed co-localization of GABA and the B cell compartment in iLNs (Extended Data Fig. 1b). In contrast to previous studies[4,5], we found a positive correlation of plasma GABA levels with disease activity scores and autoantibody titres in patients with rheumatoid arthritis, suggesting that GABA is indicative of B cell activation in humans (Fig. 1f). Together, the results indicate that GABA synthesis in LNs is enhanced by antigenic stimulation, in a B cell-dependent manner.

## B cells synthesize and secrete GABA

GABA is a major inhibitory neurotransmitter regulating inter-neuron communication. Outside the brain, GABA has been detected in the gut, spleen, liver and pancreas[6,7]. Quantitative measurements in non-immunized WT mice revealed higher amounts of GABA in peripheral and mucosal LNs than in liver or pancreas, supporting the notion

that B cell-enriched lymphoid tissues are important sources of GABA production (Extended Data Fig. 2a). However, aside from pancreatic beta cells, the cellular source of GABA outside neurons remains unknown. Although the GABA precursors glutamine and glutamate were abundant in B cells and myeloid cells, B cells were characterized by an enrichment in GABA, in either resting (contralateral) or activated (ipsilateral) popliteal LNs (Fig. 2a). B cells from bone marrow, spleen, Peyer's patches and IgA⁺ plasma cells from the small intestine lamina propria were also characterized by elevated GABA levels (Fig. 2b). Additionally, GABA and other glutamate metabolism components were relatively more abundant in B cells in non-targeted MS analyses of lymphocytes from mouse LNs or peripheral human blood (Extended Data Fig. 2b, c). Analysis of two key enzymes that convert glutamate to GABA showed that transcripts encoding glutamate decarboxylase 67 (GAD67), but not GAD65, were elevated in both mouse and human B cells compared with T cells (Extended Data Fig. 2d, e), suggesting that glutamate metabolism characterizes B-lineage cells in both species.

Next, B cells were activated in vitro in the presence of ¹³C₅,¹⁵N₂-labelled glutamine, and glutamine catabolism and labelled metabolite distribution were traced in cell lysate or supernatant in reference to stimulated T cells. Tracing indicated that glutamine was readily converted to glutamate and used for energy generation, control of reactive oxygen species (ROS) and as a source for carbon and nitrogen for building the biomass in activated B and T cells (Extended Data Fig. 3a, b). However, the abundance of labelled intracellular GABA increased in a time-dependent manner almost exclusively in B cells (Fig. 2c). Labelled glutamine-derived GABA was also detected in B cell media 72 h after stimulation, suggesting that GABA is also released from the cell (Fig. 2c). Other modes of B cell activation, including Toll-like receptor (TLR) stimulation by lipopolysaccharide (LPS) or cross-linking the B cell antigen receptor (BCR) with anti-IgM antibody alone, also induced production and secretion of GABA, albeit to a lesser extent (Fig. 2d). Various modes of stimulation of human tonsil or blood B cells also facilitated the conversion of glutamine to GABA (Fig. 2e) and increased the levels of both intracellular and secreted GABA derived from labelled glutamine (Fig. 2f and Extended Data Fig. 3c). IMS analysis of tonsil confirmed high glutamine and GABA levels in B cell follicles (Extended Data Fig. 3d). The metabolic profile of in vivo and in vitro antigen-exposed lymphocytes indicated that active glutamine and glutamate metabolism contributes to GABA production and secretion in both mouse and human B cells.

## B cells limit cytotoxic T cells via GABA

We next addressed the effect of GABA on cellular immune responses, using the MC38 colon carcinoma model in which B cells have been shown to inhibit anti-tumour T cell responses through antigen-non-specific mechanisms[8,9]. We confirmed that *muMt*⁻/⁻ mice controlled tumour growth better than their WT counterparts (Fig. 3a). However, implantation of a slow-release GABA pellet led to a significant increase in tumour growth in *muMt*⁻/⁻ mice compared with mice receiving a placebo (Fig. 3a). *muMt*⁻/⁻ tumour tissues were enriched in infiltrating CD8⁺ T cells with enhanced cytotoxic and inflammatory properties (Fig. 3b, c, and Extended Data Fig. 4a–c), which was suppressed in *muMt*⁻/⁻ mice with GABA implants (Fig. 3b, c, and Extended Data Fig. 4a–c). This phenotype was confirmed by gene expression analyses showing that upregulation of tumour necrosis factor (TNF) target gene transcripts characterized CD8⁺ T cells from *muMt*⁻/⁻ mice, and was considerably reduced by exogenous GABA treatment (Extended Data Fig. 4d). Exogenous GABA did not change tumour growth in WT mice, suggesting that endogenous GABA production (by B cells, macrophages and possibly tumour cells) saturates the system, and additional GABA cannot further impede T cell responses (Fig. 3a). However, picrotoxin, a prototypic GABAₐ receptor antagonist, limited tumour growth and enhanced the cytotoxic

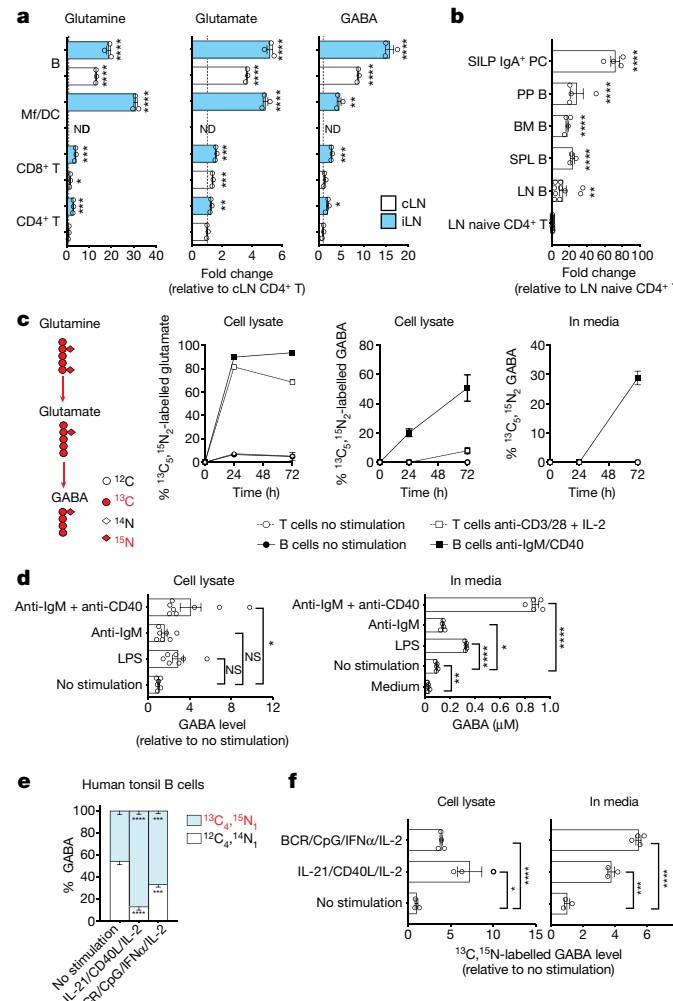

**Fig. 2 | Mouse and human B cells synthesize and secrete GABA. a**, MS analysis of glutamine, glutamate and GABA in purified CD4$^+$ T cells, CD8$^+$ T cells, CD11b$^+$ and/or CD11c$^+$ macrophages and dendritic cells (Mf/DC) or B220$^+$ B cells from the cLNs and iLNs of immunized WT mice as in Fig. 1a ($n = 3$; ND, not done). **b**, MS-determined GABA levels in B cells from LN ($n = 10$), spleen (SPL; $n = 4$), bone marrow (BM; $n = 3$), Peyer's patches (PP; $n = 4$) and small intestinal lamina propria IgA$^+$ plasma cells (SILP IgA$^+$ PC; $n = 4$), relative to naive CD4$^+$ T cells from the LN of non-immunized WT mice ($n = 13$). **c**, B cells (± anti-IgM and anti-CD40) and T cells (± anti-CD3, anti-CD28 and IL-2) were cultured with $^{13}C_5,^{15}N_2$-labelled glutamine for 24 h ($n = 2$) or 72 h ($n = 4$). Isotope-labelled glutamate and GABA were measured by MS in cell lysate or supernatant. **d**, GABA measured by MS in B cells or supernatant after 72 h of treatment with anti-IgM and anti-CD40 ($n = 8$), anti-IgM ($n = 7$) or LPS ($n = 7$), relative to non-stimulated cells ($n = 6$) or as absolute concentration ($n = 5$). **e**, **f**, Human tonsil B cells were stimulated with a mix of anti-IgM and anti-IgG (BCR), CpG (TLR9 agonists), IFNα and IL-2 ($n = 4$) or a mix of IL-21, CD40L and IL-2 ($n = 3$) for 5 d with $^{13}C_5,^{15}N_2$-labelled glutamine. The percentage of isotope-labelled and unlabelled GABA in cells (**e**) and the level of isotope-labelled GABA in cells and supernatant relative to that in non-stimulated cells (**f**), as measured by MS, are presented ($n = 3$). Significance was calculated by unpaired two-tailed $t$-test (**a**, **b**, **f**), one-way ANOVA (**d**) or two-way ANOVA (**e**): *$P < 0.05$, **$P < 0.01$, ***$P < 0.001$, ****$P < 0.0001$; NS, not significant;. Data are shown as mean ± s.e.m. (**a**–**d**, **f**) or –s.e.m. (**e**). $n$ indicates the number of biological (**a**–**c**) or technical (**d**–**f**) replicates. Data are pooled from two (**d**, left) or three (**b**) experiments. Exact $P$ values are provided in the Source Data.

activity of tumour-infiltrating CD8$^+$ T cells in WT mice (Fig. 3d, e). Neither GABA nor picrotoxin treatment affected the proliferation and viability of MC38 cells in vitro (Extended Data Fig. 4e). Together, these results indicate that reduced GABA, or GABA$_A$ signalling, enhances cytotoxic T cell responses and anti-tumour immunity, while secretion of GABA conditions the host towards immune tolerance permissive of tumour growth.

## GABA$_A$ receptors modulate CD8$^+$ T cells

Previous studies have shown that peripheral murine and human lymphocytes express functional GABA receptors[10–14]. Engagement of ionotropic GABA$_A$ receptor by GABA has been proposed to induce depolarization of membrane potential, leading to inhibition of T cell responses[10–12,15]. We examined the effect of the GABA$_A$ receptor antagonist picrotoxin on intracellular Ca$^{2+}$ concentration, and observed enhanced Ca$^{2+}$ mobilization in mouse CD8$^+$ T cells and human Jurkat T cells (Extended Data Fig. 4f). Furthermore, picrotoxin enhanced thapsigargin (TG)-induced store-operated calcium entry in mouse and human T cells (Extended Data Fig. 4g). This confirmed that functional GABA$_A$ receptors on the surface of T cells modulate a pivotal signalling pathway, confirming previous studies suggesting that GABA may directly inhibit CD8$^+$ T cells[16,17]. Purified naive CD8$^+$ T cells stimulated in the presence of GABA secreted less inflammatory cytokines, and stimulation in the presence of muscimol, a selective GABA$_A$ receptor agonist, significantly decreased activation and proliferation in a dose-dependent manner (Extended Data Fig. 4h, i). These results indicate that direct signalling via GABA$_A$ receptors is one mechanism by which GABA released by activated B cells may influence the functional properties of nearby T cells.

## GABA elicits anti-inflammatory TAMs

Tumour-associated macrophages (TAMs) are known to inhibit anti-tumour immune responses[18]. In the MC38 tumour model, macrophage depletion significantly reduced tumour growth in WT mice (Extended Data Fig. 5a, b), consistent with previous studies[19,20]. Conversely, depletion of macrophages led to an increase in tumour size in *muMt*$^{-/-}$ mice (Extended Data Fig. 5a, b), suggesting that macrophages have distinct immune-regulatory properties in the presence or absence of B cells. The TAM gene transcription profile of *muMt*$^{-/-}$ mice differed significantly from that of WT mice, but resembled that of WT mice following treatment with exogenous GABA (Extended Data Fig. 5c). Differential gene expression analyses revealed that expression of transcripts related to cytokines, particularly in the TNF signalling pathway, was enhanced in TAMs from placebo-treated *muMt*$^{-/-}$ mice compared with those of WT mice, and this phenotype was significantly disrupted when *muMt*$^{-/-}$ mice were supplemented with GABA (Fig. 3f). Upstream regulator analysis highlighted that many target genes of inflammatory cytokines such as TNF and interferon-γ (IFNγ) with increased expression in *muMt*$^{-/-}$ TAMs were also downregulated by GABA supplementation (Fig. 3g and Extended Data Fig. 5d, e). Conversely, GABA supplementation in *muMt*$^{-/-}$ mice enhanced expression of transcripts related to translation, the cell cycle and energy homeostasis such as oxidative phosphorylation (OXPHOS), which were downregulated in TAMs from *muMt*$^{-/-}$ mice compared with those from WT mice (Fig. 3f and Extended Data Fig. 5f). GABA$_A$ receptor agonists have been shown to diminish the production of inflammatory cytokines by antigen-presenting cells[21]. We found that TAMs isolated from picrotoxin-treated WT mice upregulated expression of transcripts related to calcium signalling and inflammatory cytokines, such as IFNγ-targeted genes (Fig. 3h and Extended Data Fig. 5g). Genes related to translation, the cell cycle and mitochondria and genes targeted by the interleukin-10 receptor (IL-10R) were downregulated by picrotoxin (Fig. 3h and Extended Data Fig. 5g). These results strongly suggest that GABA affects fundamental processes of macrophage physiology and facilitates polarization towards an anti-inflammatory phenotype.

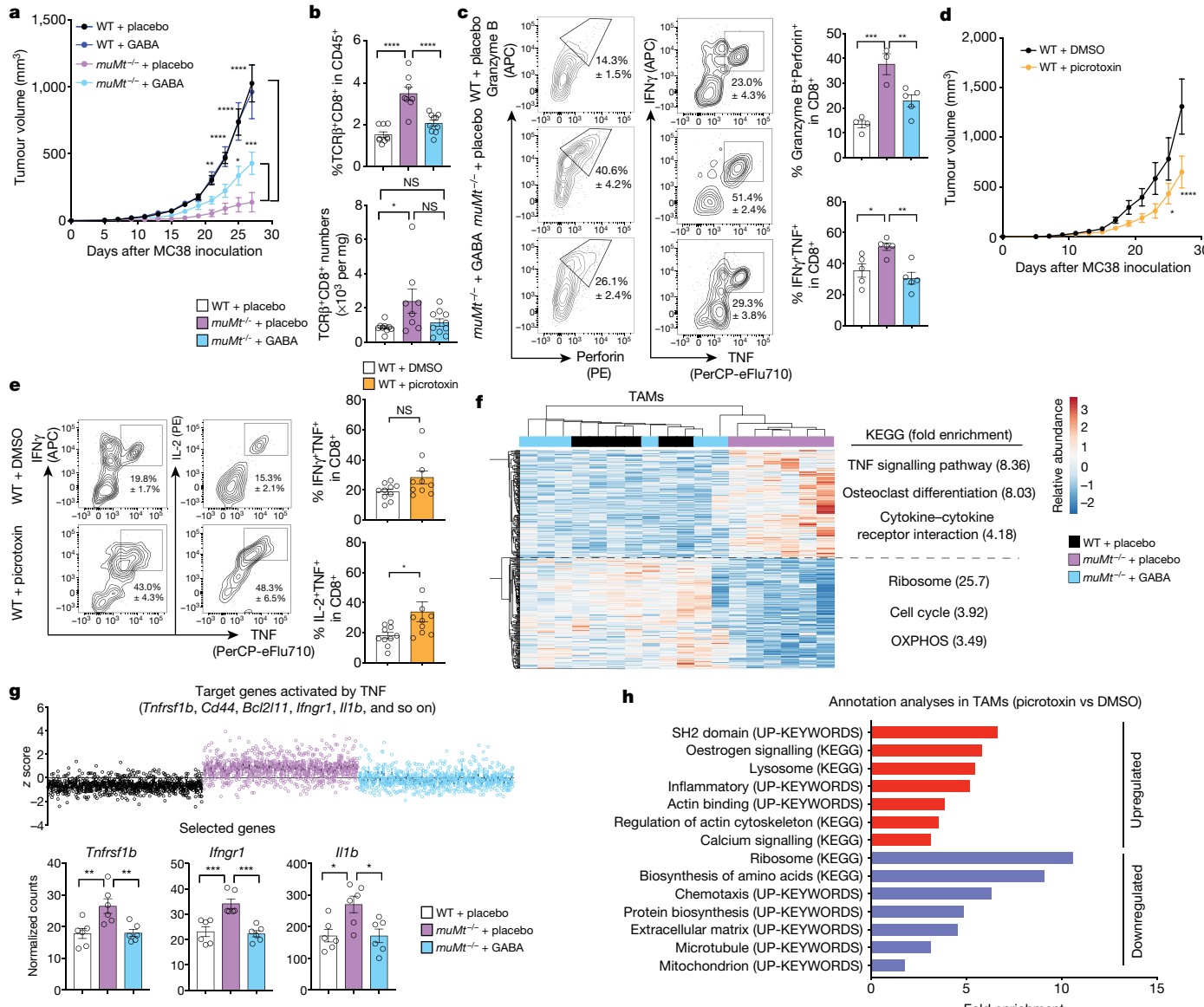

**Fig. 3 | B cells limit anti-tumour responses via GABA. a**, MC38 tumour growth in WT or *muMt*[−/−] mice implanted with GABA (WT, $n = 7$; *muMt*[−/−], $n = 8$) or placebo (WT, $n = 8$; *muMt*[−/−], $n = 6$) pellets. **b,c**, Flow cytometry quantification of tumour TCRβ[+]CD8[+] T cells (WT + placebo, $n = 9$; *muMt*[−/−] + placebo, $n = 8$; *muMt*[−/−] + GABA, $n = 10$) (**b**) and intracellular cytokines after re-stimulation (WT + placebo, $n = 4–5$; *muMt*[−/−] + placebo, $n = 3–5$; *muMt*[−/−] + GABA, $n = 5$) (**c**) on day 7 after MC38 inoculation as in **a**. **d**, MC38 tumour growth in picrotoxin- or DMSO-treated WT mice ($n = 6$). **e**, Flow cytometry of intracellular cytokines in tumour TCRβ[+]CD8[+] cells from day 7 as in **d** ($n = 10$). **f**, Hierarchical clustering and heat map of mRNA transcript abundance in TAMs purified at day 7 as in **a** ($n = 6$). Differentially expressed genes in TAMs (DEGs; two-sided Wald test, $P < 0.05$) when comparing placebo-treated WT and *muMt*[−/−] mice and *muMt*[−/−] mice treated with GABA or placebo are shown, with DEG annotation analysis

comparing the placebo-treated *muMt*[−/−] and WT groups (right). **g**, Top, *z*-scores of upregulated genes predicted to be activated by TNF in TAMs from *muMt*[−/−] mice treated with placebo compared with the WT group for all groups as in **f**; bottom, normalized number of molecules from representative genes ($n = 6$). **h**, DEG annotation analyses comparing TAMs isolated on day 7 of MC38 growth as in **d** ($n = 6$ from four biologically independent mice). Significance was calculated by two-way ANOVA (**a, d**), one-way ANOVA (**b, c, g**) or unpaired two-tailed *t*-test (**e**): *$P < 0.05$, **$P < 0.01$, ***$P < 0.001$, ****$P < 0.0001$; NS, not significant. Data are shown as mean ± s.e.m. $n$ indicates the number of biological replicates (**a–g**). Data are representative of two experiments (**a, c, d**) or pooled from two experiments (**b, e**). Exact *P* values are provided in the Source Data.

## Macrophage conditioning by GABA

Because TAMs are mostly derived from monocytes[22,23], we assessed whether GABA influenced differentiation of mouse and human monocytes into macrophages. GABA added to macrophages differentiated under neutral conditions (M-0) increased cell number, cell viability and expression of folate receptor β (FRβ), which characterizes anti-inflammatory macrophages[24] (Fig. 4a, b, and Extended Data Fig. 6a–c). Gene transcripts related to the cell cycle (such as *Mki67*,

*Ccnd1* and *Myc*) and folate metabolism (such as *Folr2*, *Mthfd2* and *Dhfr*) were upregulated by GABA (Extended Data Fig. 6d). Transcriptome and proteome profiling identified a distinct GABA M-0 signature, characterized by activation of pathways related to energy metabolism such as OXPHOS and PPAR signalling and downregulation of pathways related to neuroinflammation and nitric oxide (NO) and ROS production (Fig. 4c). Indeed, real-time measurements confirmed that GABA conditioning increased macrophage bioenergetics, particularly mitochondrial respiration (Extended Data Fig. 6e).

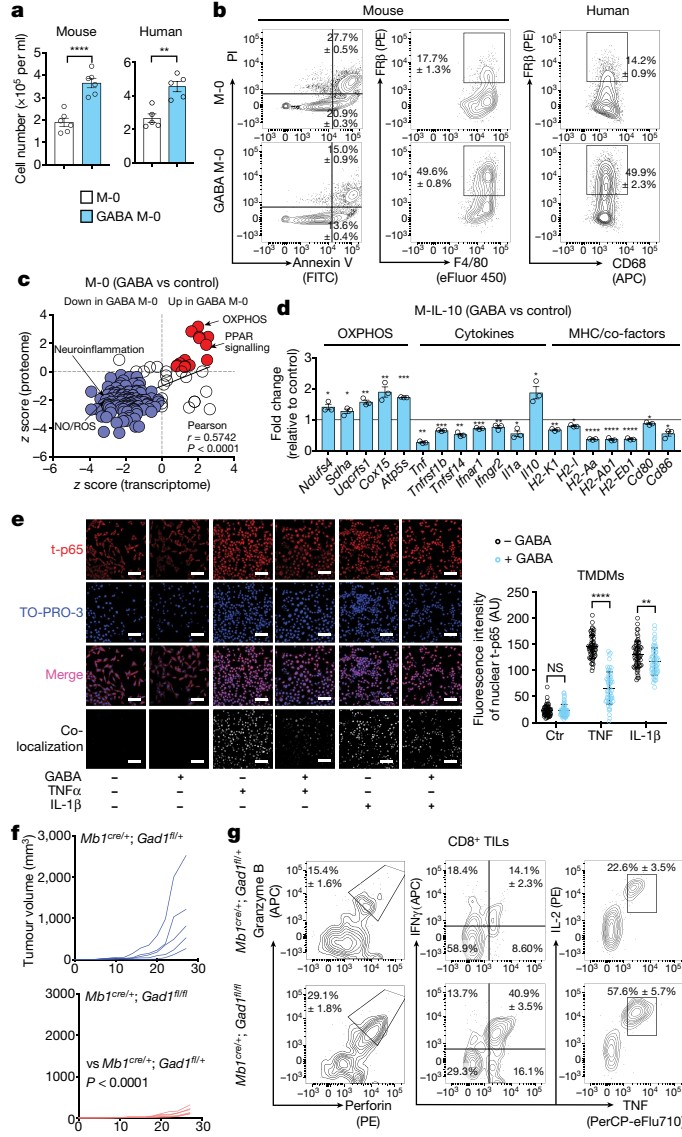

**Fig. 4 | B cell-derived GABA differentiates anti-inflammatory macrophages, promoting tumour growth. a**, Quantification of mouse bone marrow or human monocytes cultured in the presence of macrophage colony-stimulating factor (M-CSF) for 6 d (mouse, $n = 6$) or 7 d (human, $n = 5$) with (GABA M-0) or without (M-0) 1 mM GABA. **b**, Flow cytometry of GABA M-0 and M-0 cell viability (assessed with propidium iodide (PI) and Annexin V, $n = 4$), F4/80, CD68 and FRβ (mouse, $n = 3$; human, $n = 5$). **c**, Pathway analysis of mouse GABA M-0 and M-0 transcriptomes ($n = 2$) and proteomes ($n = 3$) (red, upregulated (>0); purple, downregulated (<0)). **d**, Gene expression of mouse M-0 and GABA M-0 cells after 6 h of treatment with IL-10 (M-IL-10 and GABA M-IL-10) ($n = 3$). **e**, Day 7 MC38 tumour monocytes differentiated in vitro into tumour monocyte-derived macrophages (TMDMs) with or without treatment with 1 mM GABA. Immunocytochemistry shows nuclear (TO-PRO-3) localization of total NF-κB p65 (t-p65) after TNF or IL-1β stimulation. Scale bars, 100 μm ($n = 60$). AU, arbitrary units. **f**, MC38 tumour volume in $Mb1^{cre/+};Gad1^{fl/+}$ ($n = 5$) or $Mb1^{cre/+};Gad1^{fl/fl}$ ($n = 6$) mice. **g**, Flow cytometry of intracellular cytokines in CD8$^+$ tumour-infiltrating lymphocytes (TILs) after re-stimulation ($Mb1^{cre/+};Gad1^{fl/+}$, $n = 4$; $Mb1^{cre/+};Gad1^{fl/fl}$, $n = 5$). Significance was calculated by two-tailed unpaired $t$-test (**a**, **d**) or two-way ANOVA (**e**, **f**): *$P < 0.05$, **$P < 0.01$, ***$P < 0.001$, ****$P < 0.0001$; NS, not significant. Data are shown as mean ± s.e.m. $n$ indicates the number of biological (**c**, **d**, **f**, **g**) or technical (**a**, **b**, **e**) replicates. Data are representative of two (**c** (proteome)) or three (**a** (mouse), **b** (mouse)) experiments. Exact $P$ values are provided in the source data.

Like GABA, IL-10 has also been shown to elicit anti-inflammatory macrophages by metabolic reprogramming promoting OXPHOS[25], and is known to be secreted by B cells and IgA$^+$ plasma cells[26,27]. We observed that the IL-10 receptor A signalling pathway was upregulated in TAMs from WT mice compared with $muMt^{-/-}$ mice and downregulated in WT mice by picrotoxin (Extended Data Fig. 5d, g). We next investigated how these two immune-regulatory molecules might influence macrophage generation and function. Monocytes were differentiated in the presence or absence of GABA, followed by the addition of IL-10 to cell cultures (hereafter, GABA M-IL-10 and M-IL-10), before transcriptional, bioenergetic and functional analyses (Extended Data Fig. 7a). Most transcripts of cytokines, cytokine receptors and major histocompatibility complex (MHC) presentation pathway molecules were downregulated by GABA pre-treatment, while IL-10 and OXPHOS transcripts were significantly upregulated (Fig. 4d and Extended Data Fig. 7h). GABA$_A$ receptors were involved in these transcriptomic changes, as the addition of picrotoxin partially reverted the effect of GABA on M-IL-10 cells, including IL-10 transcripts (Extended Data Fig. 7h). Real-time PCR and bioenergetic profiling confirmed that combined GABA and IL-10 increased IL-10 transcription and enhanced mitochondrial respiration, indicating generation of macrophages with anti-inflammatory properties (Extended Data Fig. 7b–d). Indeed, M-IL-10 cells conditioned with GABA or B cells significantly suppressed CD8$^+$ T cell activation in a co-culture assay, reducing granzyme B production and IFNγ and TNF secretion relative to M-IL-10 cells (Extended Data Figs. 7e, f, 8). The inhibitory effect was partially dependent on IL-10, as IL-10-blocking antibodies restored granzyme B to control levels, without affecting inflammatory cytokine production (Extended Data Figs. 7e, f, 8). In vivo, the transfer of GABA M-IL-10 cells facilitated MC38 tumour growth compared with controls receiving no cells or M-IL-10 cells (Extended Data Fig. 7g).

We next asked whether there is a specific convergence of GABA signalling on the TNF signalling pathway. Monocytes isolated from MC38 tumours were differentiated in vitro in the presence of GABA, before stimulation with TNF or IL-1β to induce nuclear factor (NF)-κB activation. GABA greatly reduced the nuclear localization of total p65 induced by TNF, while only partially attenuating its translocation induced by IL-1β (Fig. 4e). Together, these results indicate that GABA facilitates differentiation, expansion and survival of macrophages with anti-inflammatory properties.

## B cell-specific deletion of GAD67

We finally asked whether B cell-specific reduction of GABA synthesis is sufficient to restore anti-tumour responses. Mice carrying a *loxP*-flanked *Gad1* gene encoding GAD67 ($Gad1^{fl/fl}$) were crossed to transgenic mice with B cell-specific expression of Cre (*Cd79a-cre*; referred to hereafter as *Mb1-cre*) to generate $Mb1^{cre/+};Gad1^{fl/fl}$ mice. $Mb1^{cre/+};Gad1^{fl/fl}$ mice developed normally, and the frequency and number of B cells and their precursors in the bone marrow were similar to those in control $Mb1^{cre/+};Gad1^{fl/+}$ mice, as were the frequency and number of B and T cell subsets in peripheral lymphoid tissues, the gut and the peritoneal cavity (Extended Data Fig. 9). We confirmed that conditional inactivation of GAD67 in $Mb1^{cre/+};Gad1^{fl/fl}$ mice reduced GABA in B cells to levels similar to those observed in T cells (Extended Data Fig. 10a). $Mb1^{cre/+};Gad1^{fl/fl}$ mice significantly controlled growth of implanted MC38 tumours compared with $Mb1^{cre/+};Gad1^{fl/+}$ mice (Fig. 4f), with tumour tissues characterized by infiltrating CD8$^+$ T cells with enhanced cytotoxic and inflammatory properties (Fig. 4g and Extended Data Fig. 10b). Together, these results indicate that GABA produced by B cells significantly limits anti-tumour T cell responses.

## Discussion

The demands for biomass building and synthesis of effector molecules during immune activation require large adjustments in cell metabolism,

and generate countless small molecules. Small metabolites have great evolutionary potential as communication molecules, as they can be synthesized and secreted much more rapidly, using fewer cellular resources, than components of classical cell signalling pathways mediated by cytoplasmic, membrane-bound or secreted proteins. This research builds on a body of work describing GABA acting on mature immune cells, such as CD4[+] T cell effector subsets and haematopoietic precursors in the bone marrow. For many of these studies, the source of GABA remained unclear[16,28–33]. We show that GABA is produced and secreted by both mouse and human B cells, and demonstrate that B cell- or plasma cell-derived GABA is a decisive factor regulating macrophage and CD8[+] T cell responses and tumour growth in a mouse model of colon cancer.

The presence of IgA[+] plasma cells expressing PD-L1 and IL-10 within the tumour environment has been linked with poor T cell immunity in human prostatic and liver cancers[34,35]. Increased tumour-infiltrating IL-10-producing regulatory B cells also correlated with immune evasion in patients with gastric cancer[36]. We observed little infiltration of tumours by B cells or IgA[+] plasma cells in the MC38 tumour model, suggesting that B cell conditioning of CD8[+] T cells or monocytes in this model may occur upstream, during cell differentiation, migration or priming in the LN. However, in human renal cell tumours heavily infiltrated with B cells and IgA[+] plasma cells, GABA was almost exclusively detected in B cell and IgA[+] plasma cell areas (Extended Data Fig. 10c), suggesting that GABA produced in the tumour microenvironment may also regulate T cells and monocytes in some settings, perhaps explaining the poor prognosis for renal cell cancers with high infiltration of B-lineage cells[37].

The fluctuations in metabolite uptake and secretion that accompany cell activation during immune responses appear to influence both nearby cells and immune outcomes in distant organs. Understanding how intracellular metabolite networks extend into the extracellular milieu to mediate interactions between cells may facilitate development of targeted therapeutic approaches to inhibit tumour cell growth while sparing, or even enhancing, cellular immunity to cancer.

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

# Methods

## Mice

$muMt^{-/-}$ (C57BL/6J), $Cd3e^{-/-}$ (C57BL/6J)[38] $Rag1^{-/-}$ (C57BL/6J) and WT mice were bred and maintained under specific-pathogen-free conditions at the RIKEN Center for Integrative Medical Sciences. $Mb1^{cre/+}$ (C57BL/6J)[39] and $Gad1^{fl/+}$ (C57BL/6J)[40] mice were provided by M. Reth (University of Freiburg) and Y. Yanagawa (Gunma University Graduate School of Medicine), respectively. Male $Mb1^{cre/+}$; $Gad1^{fl/+}$ and female $Gad1^{fl/+}$ mice were crossed to generate mice with B cell-specific $Gad1$ knockout ($Mb1^{cre/+}$; $Gad1^{fl/fl}$) or control mice ($Mb1^{cre/+}$; $Gad1^{fl/+}$). Littermate or appropriate age- and sex-matched mice were used for analyses. All animal experiments were conducted in accordance with protocols approved by the Institutional Animal Care and Use Committee at RIKEN.

## Human tissue samples

The tonsil samples were collected from donors who (or whose parent(s), in the case of children) provided informed consent on the study, which has been approved by the IRB board at Kyoto University (IRB number: G-1250). Renal cancer samples were collected from the donors who provided informed consent on the study, which has been approved by the IRB board at Kyoto University (IRB number: G-1012).

## Foot-pad immunization

Mice were immunized with CFA-emulsified (1:1) OVA in the left foot pad (20 µg per mouse in approximately 20 µl). Seven days later, cLNs and iLNs were isolated for further analyses.

## Metabolome analysis

Metabolome analysis was performed as described previously[41]. In brief, frozen tissues or cells were homogenized in methanol, followed by addition of chloroform and ultrapure water. After centrifugation and filtration (Ultrafree-MC, UFC3 LCC NB, Human Metabolome Technologies), the solvent was removed using a vacuum concentrator (SpeedVac, Thermo). The concentrated filtrate was used for metabolite analysis. For quantification of GABA, 600 nM of 4-aminobutyric-2,2,3,3,4,4-d6 acids (Sigma-Aldrich) was dissolved in methanol before lysis of the samples. MetaboAnalystR 5.0 was used for statistical analysis (Auto scaling), enrichment analysis and metabolic pathway analysis to calculate pathway impact values (https://www.metaboanalyst.ca/home.xhtml).

## Imaging mass spectrometry

On-tissue derivatization of glutamine and GABA was performed as described previously[41]. In brief, to perform matrix-assisted laser desorption/ionization (MALDI) MS imaging of amino acids, 5 mg ml$^{-1}$ of $p$-$N$,$N$,$N$-trimethylammonioanilyl $N'$-hydroxysuccinimidyl carbamate iodide (TAHS) reagent dissolved in acetonitrile was applied to the surface of thin sections using an airbrush with a 0.2-mm nozzle calibre (Procon Boy FWA Platinum, Mr. Hobby). Tissue sections were incubated for 15 min at 55 °C, followed by application of 2,5-dihydroxybenzoic acid dissolved in acetonitrile containing 0.2% formate. IMS was performed using a MALDI ion trap mass spectrometer (MALDI LTQ XL, Thermo Scientific) equipped with a 60-Hz $N_2$ laser at 337 nm. The laser scan pitch was set at 40 µm, and the laser was irradiated 50 times for each pixel at a repetition rate of 20 Hz. Mass spectra were acquired in positive-ion mode in conjunction with consecutive reaction monitoring mode. Ion transitions at $m/z$ 323.2 > 177.1 and $m/z$ 280.2 > 177.1 (mass window, 0.75 u) were used to detect specific signals of TAHS-derivatized glutamine and GABA, respectively. Acquired data were analysed and ion images were constructed using ImageQuest (version 1.0.1, Thermo Fisher Scientific).

## Patients with rheumatoid arthritis

Patients with rheumatoid arthritis were enrolled in the Kyoto University Rheumatoid Arthritis Management Alliance (KURAMA) cohort in 2018. The SDAI and DAS28-CRP scores were used to evaluate disease severity. The levels of anti-CCP antibodies and GABA in plasma were examined. This study complied with the principles of the Declaration of Helsinki and its procedures and protocols were approved by the Medical Ethics Committee of Kyoto University Graduate School and Faculty of Medicine (approval no. R0357). Informed consent was obtained from all participants.

## Flow cytometry

Cells were stained with the following antibodies, and flow cytometry was then performed on a BD FACSAria II flow cytometry system (BD Biosciences): anti-CD8a (BioLegend, clone 53-6.7), anti-TCRβ (BioLegend, clone H57-597), anti-CD4 (BioLegend, clone RM4-5), anti-CD62L (BioLegend, clone MEL-14), anti-CD11c (BioLegend, clone N418), anti-CD11b (BioLegend, clone M1/70), anti-CD3ε (BioLegend, clone 145-2C11), anti-CD45.2 (BioLegend, clone 104), anti-granzyme B (BioLegend, clone QA16A02), anti-perforin (BioLegend, clone S16009B), anti-F4/80 (BioLegend, clone BM8), anti-cKit (BioLegend, clone 2B8), anti-SCA-1 (BioLegend, clone D7), anti-CD48 (BioLegend, clone HM48-1), anti-CD150 (BioLegend, clone TC15-12F12.2), anti-CD93 (BioLegend, clone AA4.1), anti-CD38 (BioLegend, clone 90), anti-IFNγ (eBioscience, clone XMG1.2), anti-CD44 (eBioscience, clone IM7), anti-B220 (eBioscience, clone RA3-6B2), anti-TNF (eBioscience, clone MP6-XT22), anti-IgD (eBioscience, clone 11-26c), anti-CD21/CD35 (eBioscience, clone eBio4E3), anti-FOXP3 (eBioscience, clone FJK-16s), anti-CD25 (BD Biosciences, clone PC61), anti-TCRβ (BD Biosciences, clone H57-597), anti-IL-2 (BD Biosciences, clone JES6-5H4), anti-CD23 (BD Biosciences, clone B3B4), anti-CD43 (BD Biosciences, clone S7), anti-CD16/32 (BD Biosciences, clone 2.4G2), anti-CD19 (BD Biosciences, clone 1D3), anti-CD5 (BD Biosciences, clone 53-7.3), anti-CD95 (BD Biosciences, clone Jo2), anti-γδ TCR (BD Biosciences, clone GL3), anti-IgM (SouthernBiotech, polyclonal) and anti-IgA (SouthernBiotech, polyclonal). Apoptotic cells were stained using FITC Annexin V Apoptosis Detection Kit I (BD Biosciences). Proliferating cells were analysed using the CellTrace-Violet Cell Proliferation kit (Thermo Fisher Scientific). To measure intracellular cytokine production, cells were re-stimulated with phorbol 12-myristate 13-acetate (PMA; 50 ng ml$^{-1}$) and ionomycin (500 ng ml$^{-1}$) (Sigma-Aldrich) in the presence of GolgiStop (BD Biosciences) for 4 h. Intracellular staining was performed using the Fixation/Permeabilization Solution kit (BD Biosciences). Data were analysed with FlowJo software (Tree Star).

## Cell sorting

CD4$^+$ or CD8$^+$ T cells (CD11c$^-$CD11b$^-$B220$^-$ and CD4$^+$ or CD8$^+$), naive CD4$^+$ T cells (CD11c$^-$CD11b$^-$B220$^-$CD8$^-$CD4$^+$CD44$^{low}$CD62L$^+$), B cells (CD11c$^-$ CD11b$^-$CD4$^-$CD8$^-$B220$^+$) and CD11b/c (B220$^-$CD11c$^+$ and/or CD11b$^+$) cells were sorted from LNs (pooled axillary, brachial and inguinal LNs); follicular (FO) B cells (CD19$^+$CD21$^{mid}$CD23$^+$) were sorted from spleen; Peyer's patch B cells (B220$^+$IgD$^{hi}$) and lamina propria IgA$^+$ plasma cells (B220$^-$IgA$^+$) were sorted from small intestine; and bone marrow B cells (B220$^{hi}$TCRβ$^-$) were sorted from the bone marrow of WT mice (Supplementary Fig. 1). Tumour-infiltrating CD8$^+$ T cells (CD45$^+$TCRβ$^+$CD8$^+$), monocytes (CD45$^+$CD11b$^+$F4/80$^-$Ly6C$^{hi}$) and macrophages (CD45$^+$CD11b$^+$F4/80$^{hi}$) were sorted 7 d after inoculation with MC38 cells (Supplementary Fig. 2). Fresh human peripheral blood mononuclear cells (PBMCs) were isolated by Ficoll gradient centrifugation. Fresh PBMCs, frozen PBMCs (Lonza) or frozen human tonsil cells were stained with biotin-labelled anti-CD20 (BioLegend, clone 2H7) and anti-CD19 (BioLegend, clone HIB19), and B cells were captured by magnetic selection using anti-biotin MicroBeads (Miltenyi Biotec). T cells were enriched with the Human Pan-T Cell Negative Isolation kit (Miltenyi Biotec).

## Real-time PCR

Total RNA was purified using TRIzol reagent (Invitrogen). cDNA synthesis was performed using SuperScript II Reverse Transcriptase (Invitrogen) after DNase I treatment (Invitrogen), and real-time PCR was then run

using Thunderbird SYBR Green qPCR mix (Toyobo) and results were analysed with LightCycler 96 SW 1.1 software (Roche). The relative expression levels of mRNAs were normalized to the expression of *Actb* (mouse) or 18S rRNA (human) and are represented relative to control cells. The following primers were used. Mouse primers: *Gad1* (ref. [42]): F, 5′-AGGCAGTCCTCCAAGAACCT-3′; R, 5′-CCGTTCTTAGCTGGAAGCAG-3′; *Gad2* (ref. [43]): F, 5′-TCAACTAAGTCCCACCCTAAG-3′; R, 5′-CCCTGTA GAGTCAATACCTGC-3′; *Il10* (ref. [44]): F, 5′-CAAGGAGCATTTGAATTCCC-3′; R, 5′-GGCCTTGTAGACACCTTGGTC-3′; *Actb*: F, 5′-CACCCTGTG CTGCTCACCGA-3′; R, 5′-AGTGTGGGTGACCCCGTCTCC-3′. Human primers: 18S rRNA[45]: F, 5′-GGCCCTGTAATTGGAATGAGTC-3′; R, 5′-CC AAGATCCAACTACGAGCTT-3′; *GAD1* (ref. [46]): F, 5′-CGAGTCCC TGGAGCAGAGATCCTGGTT-3′; R, 5′-GTCAGCCATTCTCCAGCTAGG CCAATAATA-3′; *GAD2* (ref. [46]): F, 5′-CAACCAAATGCATGCCTCCTACCTC TTTCA-3′; R, 5′-TGCCAACTCCAAACATTTATCAACATGCGCTTCA-3′.

## In vitro activation and [$^{13}C_5$,$^{15}N_2$]glutamine tracing

Mouse T or B cells were purified from LNs with B cell (positive selection) or T cell (negative selection) magnetic beads (Miltenyi Biotec) and cultured for 24 or 72 h in RPMI-1640 (Wako) supplemented with 10% (vol/vol) dialysed FBS (Thermo Fisher Scientific), 1× MEM NEAA, 10 mM HEPES, 50 μM of 2-mercaptoethanol, 1 mM sodium pyruvate, 100 U ml$^{-1}$ penicillin and 100 U ml$^{-1}$ streptomycin. T cells were stimulated with anti-CD3 (2.5 μg ml$^{-1}$; 145-2C11, BD Biosciences) bound to a 96-well plate in the presence of anti-CD28 (2 μg ml$^{-1}$; 37.51, BD Biosciences) and IL-2 (20 ng ml$^{-1}$; R&D Systems). B cells were stimulated with anti-IgM (8 μg ml$^{-1}$; Jackson Immuno Research) alone, anti-IgM (8 μg ml$^{-1}$) plus anti-CD40 (10 μg ml$^{-1}$; BD Biosciences) or LPS (100 ng ml$^{-1}$; Sigma-Aldrich). For [$^{13}C_5$,$^{15}N_2$]glutamine tracing, glutamine-free medium was supplemented with 2 mM of $^{13}C_5$,$^{15}N_2$-labelled L-glutamine (Taiyo Nippon Sanso).

For human B cell culture, PBMCs or tonsil-derived B cells were stimulated with a mix of human IL-21 (50 ng ml$^{-1}$; BioLegend), human CD40L (100 ng ml$^{-1}$; BioLegend) and human IL-2 (10 ng ml$^{-1}$; R&D Systems) or a mix of F(ab′)$_2$ goat anti-human IgG/IgM (1 μg ml$^{-1}$; Invitrogen), CpG oligonucleotides (ODN 2006; 4 μg ml$^{-1}$; InvivoGen), human IFNα (1,000 U ml$^{-1}$; R&D Systems) and human IL-2 (10 ng ml$^{-1}$; R&D Systems) as described previously[47] for 5 d in the medium containing $^{13}C_5$,$^{15}N_2$-labelled L-glutamine described above.

## Monocyte-derived macrophage culture

Bone marrow cells were isolated from mouse femurs. Red blood cells were removed with lysis buffer (0.15 M NH$_4$Cl, 1 mM KHCO$_3$, 0.1 mM Na$_2$EDTA). Monocytes were further purified from bone marrow cells using the EasySep Mouse Monocyte Isolation kit (STEMCELL Technologies). Total bone marrow cells (2 × 10$^6$ cells per ml) or purified monocytes (0.5 × 10$^6$ cells per ml) were suspended in complete RPMI-1640 (supplemented with 10% (vol/vol) dialysed FBS (Thermo Fisher Scientific), 1× MEM NEAA, 10 mM HEPES, 50 μM of 2-mercaptoethanol, 1 mM sodium pyruvate, 100 U ml$^{-1}$ penicillin, 100 U ml$^{-1}$ streptomycin) with 20 ng ml$^{-1}$ M-CSF (R&D Systems), and seeded on 24-well or 48-well plates. On day 3, non-adherent cells were discarded and adherent cells were further cultured for three more days with fresh medium supplemented with 20 ng ml$^{-1}$ M-CSF. Adherent cells confirmed to be CD11b$^+$F4/80$^+$ were considered to be mature BMDMs (M-0).

For polarization, M-0 cells were stimulated with 10 ng ml$^{-1}$ IL-10 (R&D Systems) for 6 h (M-IL-10). For GABA conditioning, 1 mM GABA (Sigma-Aldrich) was added from the start of the cultures, refreshed on day 3 (termed M-0 and GABA M-0) and also during the polarization period (termed M-IL-10 and GABA M-IL-10).

For culture of human monocyte-derived macrophages, human peripheral blood CD14$^+$ monocytes (Lonza) were cultured in RPMI-1640 (supplemented with 20% (vol/vol) dialysed FBS (Thermo Fisher Scientific), 1× MEM NEAA, 10 mM HEPES, 50 μM of 2-mercaptoethanol, 1 mM sodium pyruvate, 100 U ml$^{-1}$ penicillin, 100 U ml$^{-1}$ streptomycin) with 100 ng ml$^{-1}$ human M-CSF (R&D Systems) and left untreated or treated with GABA (1 mM) for 7 d. Adherent cells were analysed by flow cytometry.

## Co-culture assays

For macrophage–T cell co-cultures, 2 × 10$^4$ BMDMs differentiated in the above conditions were seeded together with 5 × 10$^4$ sorted CD8$^+$ T cells and stimulated for 3 d with anti-CD3 (1 μg ml$^{-1}$; 145-2C11, BD Biosciences) and anti-CD28 (0.5 μg ml$^{-1}$; 37.51, BD Biosciences), in the presence or absence of anti-IL-10 blocking antibodies (4 or 10 μg ml$^{-1}$; JES5-2A5, eBioscience).

For macrophage conditioning with B cells followed by T cell co-cultures, purified LN B cells were stimulated with anti-IgM (8 μg ml$^{-1}$) plus anti-CD40 (10 μg ml$^{-1}$; BD Biosciences) for 3 d. Bone marrow cells were differentiated with M-CSF for 1 d, and adherent cells were then cultured with or without activated B cells (1 × 10$^5$ cells) for five more days in the presence of M-CSF and further polarized for 6 h with IL-10 (10 ng ml$^{-1}$). Macrophage–T cell co-cultures were performed as described above. After 3 d of co-culture, CD8$^+$ T cells were analysed by flow cytometry, and the concentration of IFNγ and TNF in the supernatant was quantified using the Cytometric Bead Array (CBA) Mouse Th1/Th2/Th17 Cytokine kit (BD Biosciences).

## Imaging of NF-κB nuclear translocation

Monocytes (CD45.2$^+$CD11b$^+$Ly6C$^{hi}$F4/80$^-$) infiltrated into MC38 tumour tissues (day 7) were sorted and differentiated in vitro with or without GABA (1 mM) for 6 d and then seeded onto a 35-mm glass-bottom dish (Iwaki) and analysed by immunocytochemistry as previously described[48,49]. Cells were stimulated with recombinant mouse TNF (100 ng ml$^{-1}$) or recombinant mouse IL-1β (100 ng ml$^{-1}$) for 30 min. They were then fixed with 4% paraformaldehyde (Nacalai) for 15 min and processed for immunocytochemistry to examine the intracellular localization of total NF-κB p65. The fixed cells were permeabilized by incubation with 0.2% Triton X-100 (Sigma-Aldrich) for 15 min at 25 °C. Cells were then incubated for 90 min at room temperature with rabbit anti-human total p65 antibody (1:200; clone D14E12, Cell Signaling Technology). After washing with PBS containing 0.2% polyoxyethylene (20) sorbitan monolaurate, cells were incubated with Alexa Fluor 594-conjugated F(ab′)$_2$ fragments of goat anti-rabbit IgG (H+L) (1:1,000; Thermo Fisher Scientific) for 60 min in the dark at 25 °C. Samples were washed three times with PBS containing 0.2% polyoxyethylene (20) sorbitan monolaurate and incubated with TO-PRO-3-iodide (1:1,000; Thermo Fisher Scientific) for 15 min. They were then mounted with ProLong Gold Antifade reagent and visualized using a confocal laser scanning microscope (LSM-510, Carl Zeiss). Co-localization analysis was performed using ZEN software (Carl Zeiss).

## Immunofluorescence analysis

For immunofluorescence, tissues were immediately isolated, fixed for 2 h with 4% paraformaldehyde in PBS at 4 °C and soaked in 30% sucrose in PBS overnight at 4 °C. Tissues were then embedded in Tissue-Tek OCT blocks (Sakura) and frozen in liquid nitrogen. The frozen samples were sectioned at a thickness of 10 μm by cryostat (Leica, CM3050S). The following antibodies were used: anti-mouse CD3ε (BioLegend, clone 145-2C11), anti-mouse B220 (eBioscience, clone RA3-6B2) and anti-mouse CD11c (SouthernBiotech, polyclonal). Tissue sections were counterstained with DAPI (Sigma-Aldrich) and mounted with Fluoromount-G antifade reagent (SouthernBiotech).

For human tissue staining, tissues were freshly isolated, embedded in Tissue-Tek OCT blocks (Sakura) or SCEM (SECTION-LAB) and frozen in liquid nitrogen. The frozen samples were sectioned at a thickness of 10 μm by cryostat (Leica, CM3050S). The following antibodies were used for immunohistochemistry staining: anti-human CD68 (eBioscience, clone 815CU17), anti-human CD19 (Abcam, clone EPR5906) and anti-human IgA (SouthernBiotech, polyclonal). Fluorescence images were obtained using a BZ-X700 fluorescence microscope (Keyence).

## Extracellular flux analysis

BMDMs differentiated as described above were seeded on Seahorse XF poly(D-lysine)-coated microplates (Agilent) at $1 \times 10^5$ cells per well in Seahorse XFp RPMI medium containing 1 mM XFp sodium pyruvate, 2 mM XFp L-glutamine and 10 mM XFp glucose (Agilent) and incubated for 45 min at 37 °C in a non-$CO_2$ incubator before starting the assay using a Seahorse XFp analyser (Agilent). Oligomycin (1.5 μM), FCCP (2 μM) and rotenone/antimycin A (0.5 μM) were added sequentially, and the oxygen consumption rate (OCR) and extracellular acidification rate (ECAR) were measured in real time. The maximal respiratory capacity was calculated as (maximum OCR after FCCP injection) − (OCR after rotenone/antimycin A injection).

## Measurement of intracellular $Ca^{2+}$ concentrations

Cells were incubated with 2 μM Fluo-3-AM in RPMI supplemented with 10% FBS for 30 min at 37 °C in the dark, washed in PBS and seeded on 35-mm glass-base dishes with $Ca^{2+}$ imaging buffer (120 mM NaCl, 5 mM KCl, 0.96 mM $NaH_2PO_4$, 1 mM $MgCl_2$, 11.1 mM glucose, 1 mM $CaCl_2$, 1 mg ml$^{-1}$ BSA and 10 mM HEPES (pH 7.4)) or $Ca^{2+}$-free imaging buffer (120 mM NaCl, 5 mM KCl, 0.96 mM $NaH_2PO_4$, 1 mM $MgCl_2$, 11.1 mM glucose, 0.5 mM EGTA, 1 mg ml$^{-1}$ BSA and 10 mM HEPES (pH 7.4)). A confocal laser scanning microscope (LSM510) was used to capture images every 1 s. After baseline image acquisition, cells were stimulated with anti-CD3/CD28 Dynabeads (Thermo Fisher Scientific) or 5 μM thapsigargin (TG). To examine the effects of antagonists for the GABAergic receptor, cells were pre-treated with 100 μM picrotoxin for 5 min. The relative change in intracellular $Ca^{2+}$ concentration ($n = 60$ cells) over time is expressed as the change relative to baseline fluorescence.

## Tumour model

The MC38 (mouse colon adenocarcinoma) cell line was originally provided by J. P. Allison (Memorial Sloan Kettering Cancer Center). Mice were implanted subcutaneously with placebo or GABA pellets designed to release GABA over 21 d (31.5 mg per pellet; Innovative Research) or injected intraperitoneally (i.p.) with DMSO or picrotoxin (40 μg per mouse; Abcam) 1 d before intradermal inoculation with $5 \times 10^5$ tumour cells in the right flank. DMSO or picrotoxin injection in 200 μl of saline was performed every other day. For macrophage depletion, control liposomes or liposomal clodronate (Hygieia Bioscience) was injected into mice i.p. 1 d before (50 μl per mouse) and on day 6 after (25 μl per mouse) inoculation. For co-injection experiments, MC38 cells ($5 \times 10^5$ cells per mouse) were injected into mice together with M-IL-10 or GABA M-IL-10 cells ($2.5 \times 10^5$ cells per mouse) generated in vitro as described above. Tumour tissues were collected on day 7 or day 15 and digested with collagenase (1.5 mg ml$^{-1}$; Wako) for flow cytometry or sequencing analysis. Tumour volumes were measured with calipers according to the following formula: tumour volume = $\pi \times$ (length × breadth × height)/6.

## RNA sequencing

One hundred intratumoural CD45.2$^+$TCRβ$^+$CD8$^+$ T cells and CD45.2$^+$CD11b$^+$F4/80$^{hi}$ macrophages were purified on day 7 after tumour injection using a FACSAria III Cell Sorter (BD Biosciences). Cells were collected in SingleCellProtect Single Cell Stabilizing Solution (Avidin) and frozen in liquid nitrogen for digital RNA sequencing[50]. Each library (one per mouse) was sequenced using an indexed pooling method on the MiSeq platform (150 cycles; Illumina kit). Sequencing data were mapped to the mouse genome (mm10 assembly from the UCSC Genome Browser; annotation refFlat from the UCSC Genome Browser) using STAR v.2.5.4b[50]. The normalized number of molecules was calculated using DESeq2 (1.30.1). Genes with significantly different expression (DEGs; two-sided Wald test, $P < 0.05$; average value of normalized number of molecules for all samples >1; log$_2$(fold change) >0 or <0) were analysed by Ingenuity Pathway Analysis (IPA; Qiagen). DAVID was used for annotation analysis of DEGs (https://david.ncifcrf.gov/home.jsp).

Principal-component analysis (average value of normalized number of molecules of all samples >1) was performed through ClustVis (https://biit.cs.ut.ee/clustvis/). Sequencing datasets have been deposited in the Gene Expression Omnibus under accession codes GSE169543 and GSE183246.

## Gene chip assays

Total cellular RNA was purified using TRIzol (Invitrogen), and the quality was confirmed using Agilent RNA 6000 Pico reagent (Agilent Technologies). The Clariom S Array (Thermo Fisher Scientific) was used to analyse the transcriptome profile (Takara), and data were analysed with Transcriptome Analysis Console Software (Thermo Fisher Scientific) (Supplementary Tables 1 and 2). Pathway analysis was performed using IPA (Qiagen).

## Proteome analysis

For preparation of tryptic peptide samples for MS, the phase transfer surfactant method of refs. [51,52] was used to prepare BMDM samples for MS, with minor modifications. In brief, cell pellets of 100,000 cells were lysed in 20 μl lysis buffer (100 mM Tris-Cl pH 9.0, 12 mM sodium deoxycholate, 12 mM sodium N-dodecanoylsarcosinate, with cOmplete EDTA-free protease inhibitor (Roche)). The protein concentration of each sample was determined using a Pierce BCA Protein Assay kit (Thermo Fisher Scientific). Protein amount varied from sample to sample (3.2–10.0 μg), and the whole amount of each sample was prepared for MS. Samples were reduced with 10 mM DL-dithiothreitol at 50 °C for 30 min, free thiol groups were alkylated with 40 mM iodoacetamide in the dark at room temperature for 30 min and 55 mM cysteine was then added at room temperature for 10 min to quench the reactions. Samples were diluted 1:2.76 with 50 mM ammonium bicarbonate. Lysyl endopeptidase (FUJI-FILM Wako Pure Chemical) and modified trypsin (Promega) were both added at 200 ng, and proteins were digested at 37 °C for 14 h. Tryptic digestion reactions were treated with 1.83 volumes of ethyl acetate and then acidified with trifluoroacetic acid (TFA) to 0.5% (vol/vol) TFA. After centrifugation at 12,000$g$ for 5 min, the upper organic phase containing detergent was discarded and a lower aqueous phase containing digested tryptic peptides was dried using a vacuum centrifuge. The samples were desalted with ZipTip Pipette Tips with 0.6 μl of C18 resin (MilliporeSigma) and dried. Afterwards, samples were dissolved in 10 μl of 0.1% (vol/vol) formic acid and 3% (vol/vol) acetonitrile in water. The peptide concentrations were determined using a Pierce Quantitative Colorimetric Peptide Assay kit (Thermo Fisher Scientific). From each sample, 600 ng of tryptic peptides was measured with MS.

To generate a spectral library, aliquots of tryptic peptides from each sample were combined for a 10-μg tryptic peptide sample, which was fractionated using a Pierce High-pH Reversed-Phase Peptide Fractionation (HPRP) kit, according to the manufacturer's instructions (Thermo Fisher Scientific). For MS, each fraction was dissolved in 6.5 μl of 0.1% (vol/vol) formic acid and 3% (vol/vol) acetonitrile in water and 5.0 μl was measured.

Liquid chromatography and tandem MS (LC–MS/MS) measurements were made using a Q-Exactive Plus Orbitrap mass spectrometer together with a Nanospray Flex ion source (Thermo Fisher Scientific). For LC, an EASY-nLC 1200 system was used (Thermo Fisher Scientific). Solvent A consisted of MS-grade 0.1% (vol/vol) formic acid in water and solvent B consisted of 0.1% (vol/vol) formic acid in 80% (vol/vol) acetonitrile. Samples were measured with a 2-h gradient and a flow rate of 300 nl min$^{-1}$: the gradient increased from 2% solvent B to 34% solvent B from 0–108 min, then from 34% solvent B to 95% solvent B from 108–110 min and finally to 95% solvent B from 110–120 min to wash the system. Peptides were separated using an analytical column with 3-μm C18 particles, an inner diameter of 75 μm and a length of 12.5 cm (Nikkyo Technos), which was preceded by an Acclaim PepMap 100 trap column with 3-μm C18 particles, an inner diameter of 75 μm and a filling length of 2 cm (Thermo Fisher Scientific). The ion transfer capillary temperature was 250 °C and a spray voltage of 2.0 kV was applied during sample measurement.

To generate a spectral library, the HPRP-fractionated samples were measured with data-dependent acquisition (DDA). Full MS spectra were acquired from 380 to 1,500 $m/z$ at 70,000 resolution. The automatic gain control (AGC) target was $3 \times 10^6$, and the maximum injection time (IT) was 100 ms. $MS^2$ scans were recorded for the top 20 precursors at 17,500 resolution, and the dynamic exclusion was set to 20 s with the default charge state set to 2. The AGC target was $1 \times 10^5$ with a 60-ms maximum IT. The normalized collision energy was 27% for HCD fragmentation. The intensity threshold was $1.3 \times 10^4$, and charge states 2–5 were included.

To quantify proteins across samples, they were measured with data-independent acquisition (DIA). Data were acquired with one full MS scan and 32 overlapping isolation windows covering the precursor mass range of 400–1,200 $m/z$. For the full MS scan, the resolution was 70,000, $5 \times 10^6$ was the AGC target and the maximum IT was set to 120 ms. DIA segments were acquired at a resolution of 35,000, a $3 \times 10^6$ AGC target and an automatic maximum IT. The first mass was fixed at 150 $m/z$. The normalized collision energy was 27% for HCD fragmentation.

For generation of the spectral library, the eight raw data files obtained from DDA of the HPRP-fractionated tryptic peptides fractions were analysed using Proteome Discoverer version 2.4 (Thermo Fisher Scientific) together with eight raw data files obtained from DDA of eight HPRP-fractionated tryptic peptide fractions derived from BMDM samples, which we had measured previously. The UniProt-reviewed *Mus musculus* (taxon 10090) database was used to process the data files to generate a single results file. Digestion enzyme specificity was set to trypsin (full). Mass tolerance for the precursors was set to 10 p.p.m., and for the fragments the tolerance was 0.02 Da. Carbamidomethylation of cysteine was set as a static modification. Methionine oxidation, N-terminal protein acetylation, methionine loss and methionine loss with N-terminal protein acetylation were set as dynamic modifications. To calculate the false discovery rate (FDR), a concatenated decoy database was used. At both the peptide and protein levels, an FDR of 0.01 was used to filter the search results. To generate the specific spectral library, the results file from Proteome Discoverer was then imported into Spectronaut software (Biognosys).

For quantification of proteins, raw data files from DIA measurements were used to extract protein quantities with the generated spectral library using Spectronaut software (Biognosys). FDR was estimated with the mProphet approach[53], and set to 0.01 at both the peptide precursor and protein level[54]. Data filtering parameters for quantification were a $q$-value percentile fraction of 0.5 with global imputing and cross-run normalization with global normalization on the median. Pathway analysis based on IPA (Qiagen) was performed to analyse significantly differentially expressed proteins (two-tailed unpaired $t$-test, $P < 0.05$). The MS proteomics data have been deposited to the ProteomeXchange Consortium via the PRIDE[55] partner repository with the dataset identifier PXD028403.

## Statistical analysis

Statistical analysis was performed using Prism (GraphPad) or DESeq2. Analyses were conducted using the Pearson's and Spearman's correlation test, the Wald test, the two-tailed unpaired Student's $t$-test or repeated-measures ANOVA. $P$ value less than 0.05 was considered statistically significant.

## Reporting summary

Further information on research design is available in the Nature Research Reporting Summary linked to this paper.

## Data availability

The RNA-seq datasets analysed are publicly available in the Gene Expression Omnibus with accession codes GSE169543 and GSE183246. The proteomics datasets are available via ProteomeXchange with identifier PXD028403. The gene chip datasets are provided in Supplementary Tables 1 and 2. Source data are provided with this paper.

## Code availability

The DESeq2 (1.30.1) package was used to analyse RNA-seq data (https://bioconductor.org/packages/release/bioc/html/DESeq2.html).

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

**Acknowledgements** We thank all members of the Fagarasan laboratory for discussions. We thank Y. Yanagawa (Gunma University) for *Gad1*fl/+ mice; M. Reth (University of Freiburg) and T. Kurosaki (IMS RIKEN) for *Mb1*cre/+ mice; T. Kitami (IMS RIKEN) for the use of the Seahorse Analyser and assistance with Seahorse assays; T. Saito (IMS RIKEN) for the Jurkat cell line; the Iskikawa laboratory (IMS RIKEN) for advice on human tissue staining; M. Li (IMS RIKEN) for help with macrophage cultures; K. Fukuhara (IMS RIKEN) for RNA-seq; and S. Kato for her administrative assistance. This work was supported by the Japan Agency for Medical Research and Development–Core Research for Evolutional Science and Technology (grant 14532135 to S.F.), the Japan Agency for Medical Research and Development-FORCE (grant JP21gm4010008 to S.F.), the RIKEN Aging Project Grant (to S.F.), the Japan Agency for Medical Research and Development–Precursory Research for Innovative Medical Care (JP21gm6110019 to M. Miyajima), the Japan Society for the Promotion of Science KAKENHI Grant-in-Aid for Challenging Research (Exploratory) (grant 21K19392 to M. Miyajima), the Japan Society for the Promotion of Science KAKENHI Grant 18H05411 (K.S.) and the Yanai Fund (Kyoto University); B.Z., A.V., N.H. and T.O. were supported by the RIKEN Special Postdoctoral Researcher (SPDR) Program. The KURAMA cohort study is supported by a grant from Daiichi Sankyo. The funder had no role in the design of the study, the collection or analysis of the data, the writing of the manuscript or the decision to submit the manuscript for publication.

**Author contributions** S.F., B.Z. and A.V. designed the study. B.Z., A.V. and M. Miyajima carried out in vivo and in vitro cell characterization in immunization and tumour experiments. Y.S. and K. Sonomura performed metabolomic studies under the supervision of M.S. and F.M. Y.S. performed tracing experiments and IMS studies on mouse and human tissues. Y.W. performed proteomic studies. K.C., R.J.M. and R.H. contributed to tumour experiments and human blood cell sorting and stimulation under the supervision of T. Honjo. N.H., T.O. and K. Shiroguchi performed 100-cell transcriptome analyses. R.N. performed experiments related to calcium mobilization and NF-κB activation under the supervision of H.S. W.K., Y.T., S.Y., M. Maruya, S.N. and K. Suzuki provided technical help with mouse generation, phenotyping, analyses and

metabolome and transcriptome sample preparation. K.M. and M.H. provided samples and data for patients with rheumatoid arthritis. H.U. provided tonsil samples, and T.K., K.I. and T. Hirano provided renal tumour samples. S.F. supervised the work and analysed data, and S.F., A.V. and B.Z. wrote the manuscript.

**Competing interests** S.F., B.Z. and M. Miyajima, through RIKEN Innovation, have filed a patent application on the methods and findings in this manuscript. The remaining authors declare no competing interests.

**Additional information**
**Correspondence and requests for materials** should be addressed to Sidonia Fagarasan.

**a**

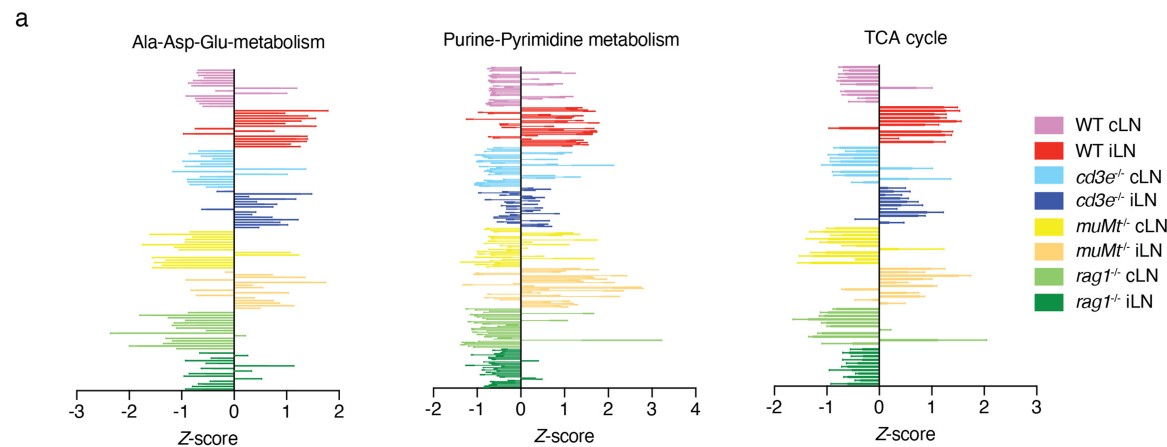

**b**

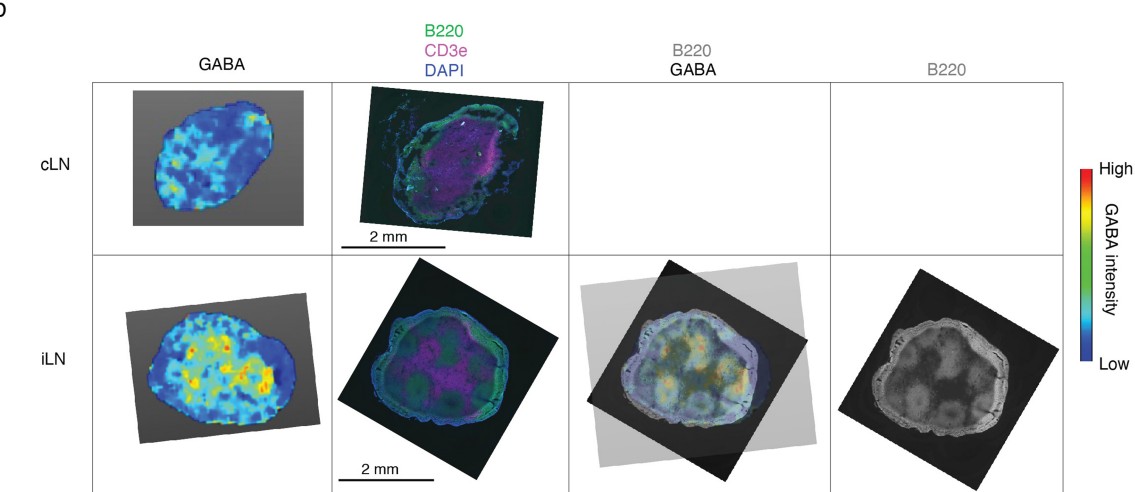

**c**

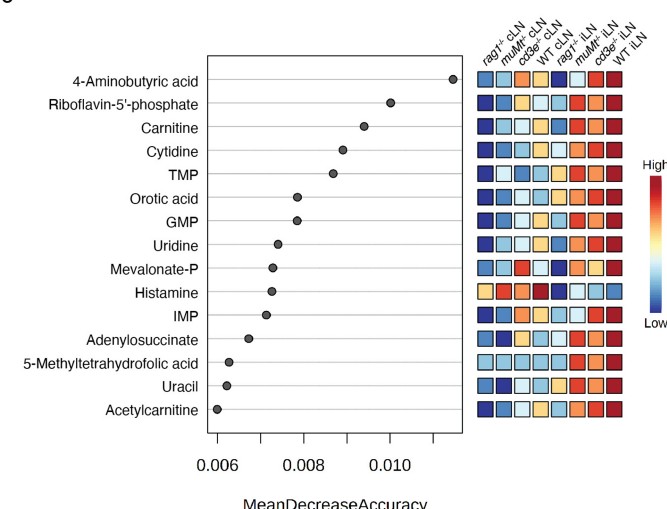

**Extended Data Fig. 1 | Metabolic pathways and top metabolites changed by immunization in WT and immunodeficient mice. a**, Pathway analyses of significantly different metabolites (two-tailed unpaired t-test *P* value < 0.05) between iLN and cLN of WT mice revealed major activation-induced changes in metabolites related to alanine-aspartate-glutamate metabolism, purine-pyrimidine metabolism and TCA cycle. The *z*-score of all detected metabolites in each of these pathways were plotted to provide an overview of metabolic similarity and variation among different groups of mice. **b**, Imaging mass spectrometry of GABA and immunohistochemical analyses of cLN and iLN of WT mice. **c**, Random forest analysis based on of the metabolites extracted from iLN and cLN of WT mice (*n* = 11), *cd3*[-/-] (*n* = 5), *muMt*[-/-] (*n* = 4) and *rag1*[-/-] (*n* = 3) mice measured by mass spectrometry (MS). *n* indicates biological replicates. Data are representative of two experiments.

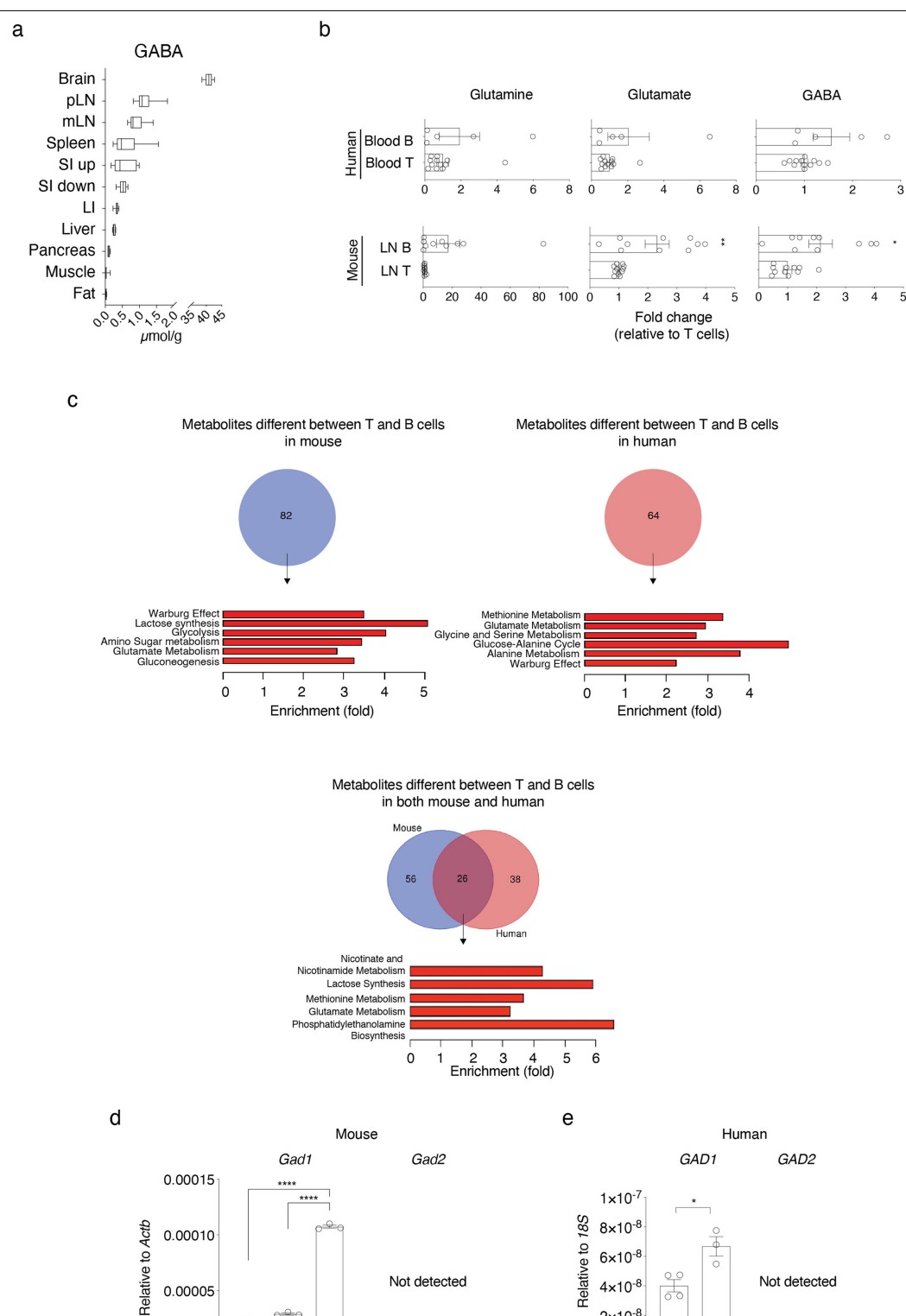

**Extended Data Fig. 2** | See next page for caption.

**Extended Data Fig. 2 | GABA concentration in multiple tissues and metabolic signature of B cells in mouse and human. a**, Quantification of GABA in brain, peripheral lymph nodes (pLN) ($n = 6$), mesenteric LN (mLN) ($n = 6$), spleen ($n = 6$), upper and lower segments of the small intestine (SI up and SI low, respectively) ($n = 6$), large intestine (LI) ($n = 6$), liver ($n = 6$), pancreas ($n = 6$), hindlimb skeletal muscle ($n = 6$) and perigonadal fat ($n = 5$) of WT mice measured by mass spectrometry. Box-and-whiskers plots represent the range from the 25th to 75th percentiles (box), the median value (the middle line) and the minimum to maximum value (whiskers). $n$ indicates biological replicates. **b**, Measurement of glutamine, glutamate and GABA by mass spectrometry in *ex vivo*-sorted T or B cells purified by FACS from mouse peripheral lymph nodes and human peripheral blood, relative to matched T cells. **c**, Venn diagrams (upper) and metabolite-set–enrichment analysis (lower panels) showing the significantly different metabolites (two-tailed unpaired t-test $P$ value < 0.05) between *ex vivo*-sorted T and B cells from mice or humans ($n = 5$ (B cells) or 12 (T cells) replicates derived from 5 biologically independent human healthy donors, $n = 10$ or 11 (glutamate and GABA in T cells) replicates derived from 6 biologically independent mice). **d, e**, qPCR analysis of GAD67 (*Gad1/GAD1*) and GAD65 (*Gad2/GAD2*) mRNA level in mouse LN-derived T and B cells ($n = 3$) (**d**), or human blood-derived T ($n = 4$) and B cells ($n = 3$) (**e**). *$P < 0.05$, ****$P < 0.0001$ (two-tailed unpaired t-test (**b, d, e**)). Bars represent mean ±SEM. Data are pooled from four experiments (**a (mouse)**) or representative of two experiments (**d**). Exact $P$ values are in Source Data.

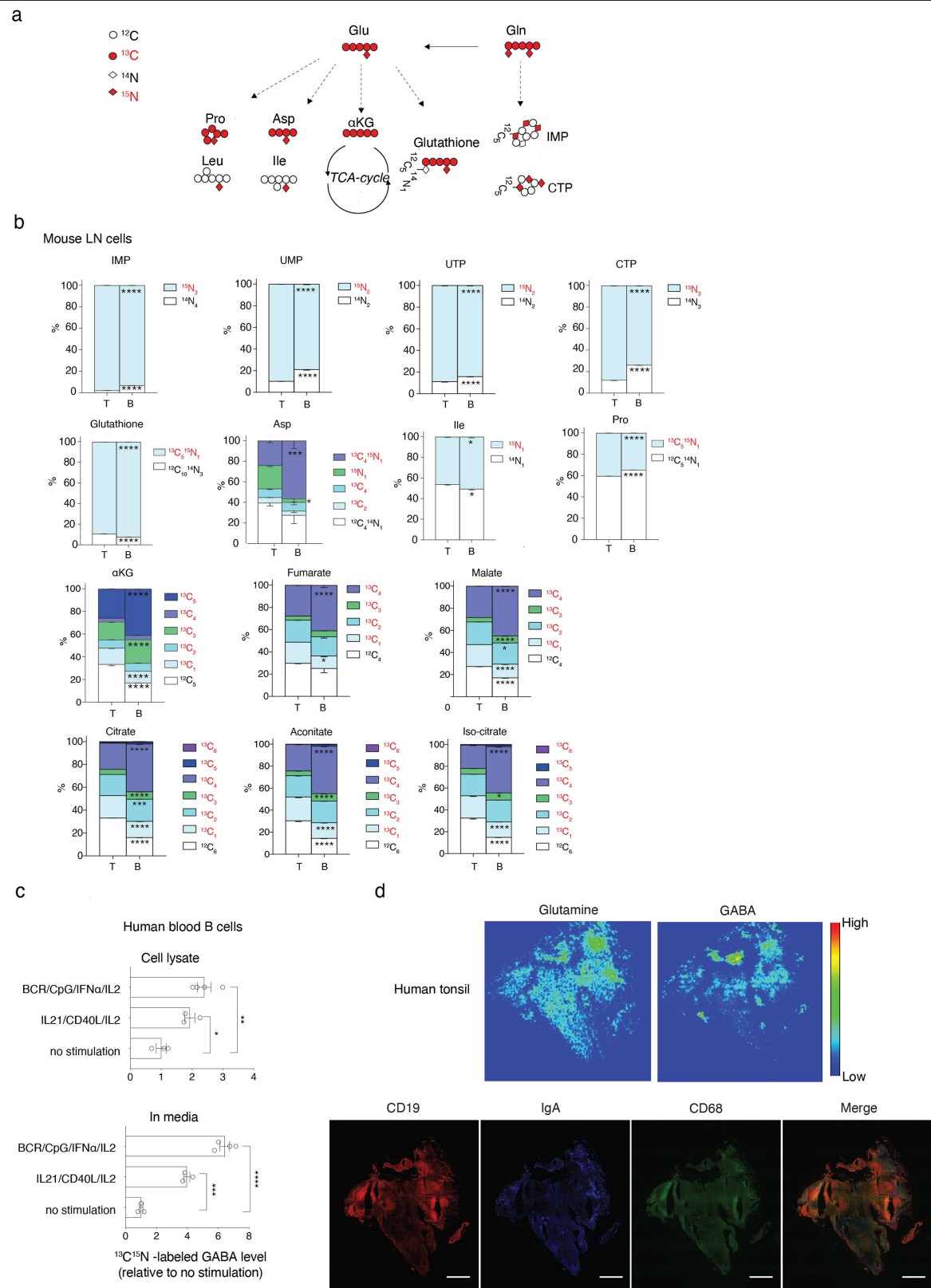

**Extended Data Fig. 3** | See next page for caption.

**Extended Data Fig. 3 | $^{13}C^{15}N$-glutamine tracing in mouse and human T and B cells *in vitro*.** Mouse T or B cells sorted from LN were cultured in medium containing $^{13}C_5^{15}N_2$-labeled glutamine (Gln) for 72 h, with or without anti-CD3, anti-CD28 and IL-2 (for T cells), or anti-IgM and anti-CD40 antibodies (for B cells) stimulation. Isotope-labeled or unlabeled metabolites in cells or supernatants were measured by mass spectrometry. **a**, Scheme of metabolism of $^{13}C_5^{15}N_2$-labeled Gln. **b**, Isotopomer analysis of isotope-labeled fraction of the indicated metabolites associated with Gln metabolism in T ($n = 3$) or B cells ($n = 4$) 72 h after stimulation. Inosine monophosphate (IMP), uridine monophosphate (UMP), uridine triphosphate (UTP), cytidine-5′-triphosphate (CTP), aspartate (Asp), isoleucine (Ile), proline (Pro), and tricarboxylic acid cycle (TCA-cycle) related metabolites- alpha-ketoglutarate (αKG), fumarate, malate, citrate, aconitate and isocitrate- are shown. The results are represented as the ratio of isotope-labeled-metabolite in total amount of each metabolite. **c**, Human peripheral blood-derived B cells were stimulated with a mix of anti-IgM/IgG (BCR), CpG (TLR9 agonists), IFN-α A and IL-2 ($n = 4$) or a mix of IL-21, CD40L and IL-2 ($n = 3$) for 5 days in medium containing $^{13}C_5^{15}N_2$-labeled Gln. The level of isotope-labeled-GABA in cell lysate or media is shown, presented relative to non-stimulated cells ($n = 3$). **d**, Imaging mass spectrometry of GABA, glutamine (upper) and immunohistochemical analyses (lower) of human tonsil tissue sections. Scale bar, 1 mm. *$P < 0.05$, ***$P < 0.001$, ****$P < 0.0001$ (two-way ANOVA (b); unpaired two-tailed t-test (c)). Bars represent mean -SEM (b) or +SEM (c). $n$ indicates biological (b) or technical replicates (c). The experiment is performed once (d). Exact $P$ values are in Source Data.

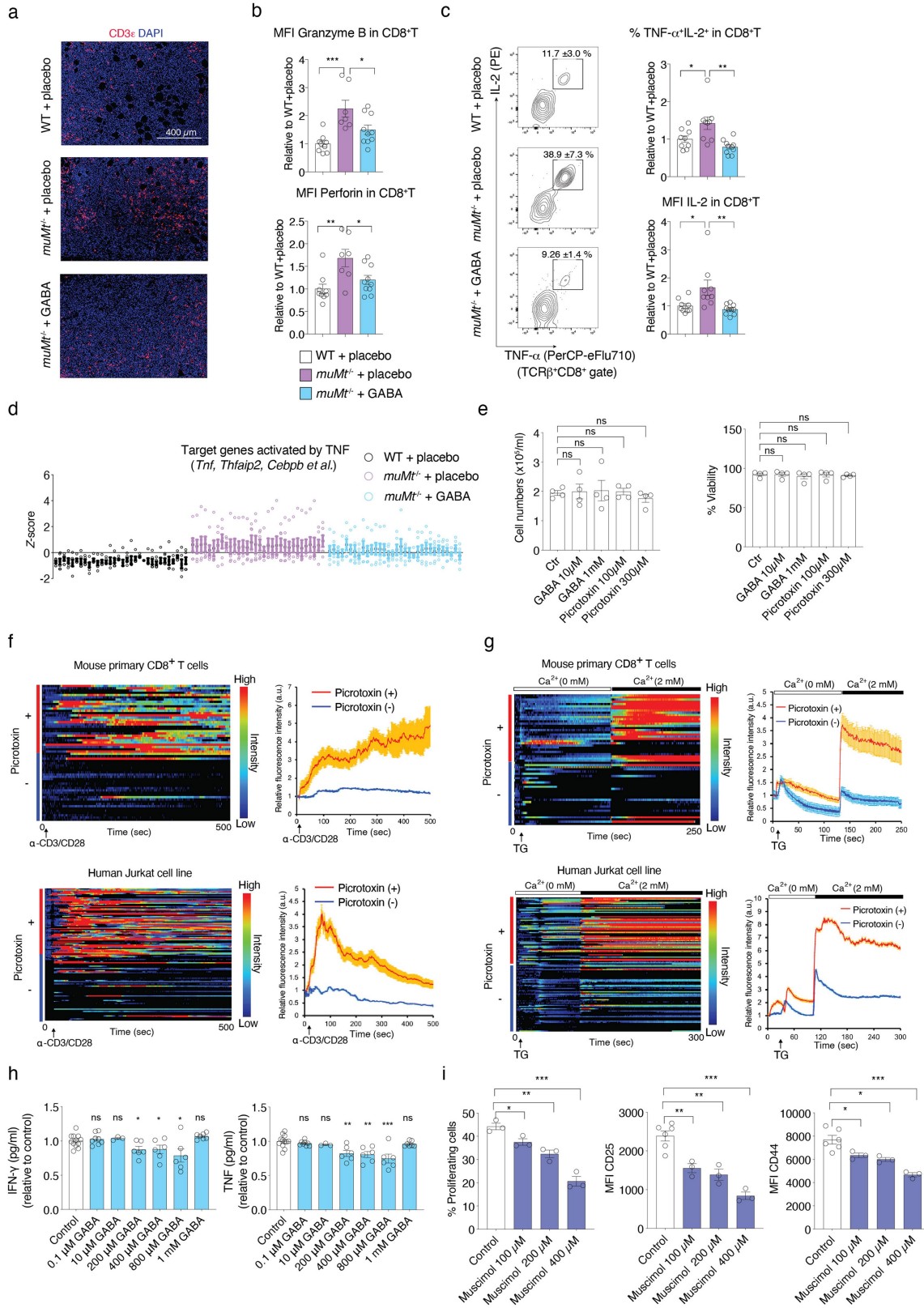

**Extended Data Fig. 4** | See next page for caption.

**Extended Data Fig. 4 | Functional GABA$_A$ receptors expressed on CD8$^+$ T cells.** WT or $muMt^{-/-}$ mice implanted with a pellet containing GABA or a placebo were injected with MC38 tumor cells as in Fig.3a. **a**, Representative immunohistochemistry of tumor sections from the indicated mice groups. **b**, The mean fluorescence intensity (MFI) of granzyme B and perforin in CD8$^+$ TILs normalized to the mean WT + placebo value (WT, $n = 9$; $muMt^{-/-}$, $n = 7$; $muMt^{-/-}$ +GABA, $n = 10$). **c**, Representative flow cytometry profile of CD8$^+$ TILs cells stained as indicated and quantification of the frequencies of TNF-α$^+$ IL-2$^+$ or MFI of IL-2 in CD8$^+$ TILs, normalized to the mean WT + placebo value (WT, $n = 9$; $muMt^{-/-}$, $n = 9$; $muMt^{-/-}$ +GABA, $n = 10$). **d**, The $z$-score of the transcripts from RNA sequencing of CD8$^+$ TILs in each group, all upregulated genes predictively activated by TNF in CD8$^+$ TILs of $muMt^{-/-}$ + placebo compared to WT group (Wald test $P < 0.05$; two-sided) are selected and plotted for all groups ($n = 6$). **e**, MC38 cells were cultured with GABA or picrotoxin as indicated for 3 days. The number and viability of cells were measured ($n = 4$). **f**, Calcium measurements of mouse CD8$^+$ T cells (upper) ($n = 30$) or human Jurkat cell line (lower) ($n = 60$) with and without picrotoxin treatment and stimulated with anti-CD3/CD28 dynabeads. **g**, The analysis of thapsigargin (TG)-induced calcium level of mouse CD8$^+$ T cells (upper) ($n = 30$) or human Jurkat cell line (lower) ($n = 60$) with and without picrotoxin treatment in the calcium-free or sufficient medium. **h, i**, Purified naïve CD8$^+$ T cells labeled with CellTrace violet dye were cultured for 3 days with or without GABA or muscimol and stimulated with anti-CD3 (1 μg/ml) and anti-CD28 (0.5 μg/ml). The concentration of IFNγ and TNF in supernatants was measured by cytometric bead array, presented as relative to control ($n = 12$ (control), 8 (0.1 μM), 3 (10 μM), 6 (200, 400, 800 μM), 7 (1 mM)) (**h**). Representative quantification analysis for the frequencies of proliferating cells, the MFI of CD25 or CD44 ($n = 3$) (**i**). ns: not significant, *$P < 0.05$, **$P < 0.01$, ***$P < 0.001$ (one-way ANOVA (**b, c**); two-tailed unpaired t-test (**e, h, i**). Bars represent mean ±SEM. $n$ indicates biological replicates (**a-d**) or technical replicates (**e-i**). Data are representative of two experiments (**a, e, f, g, i**) or pooled from two experiments (**b, c, h**). Exact $P$ values are in Source Data.

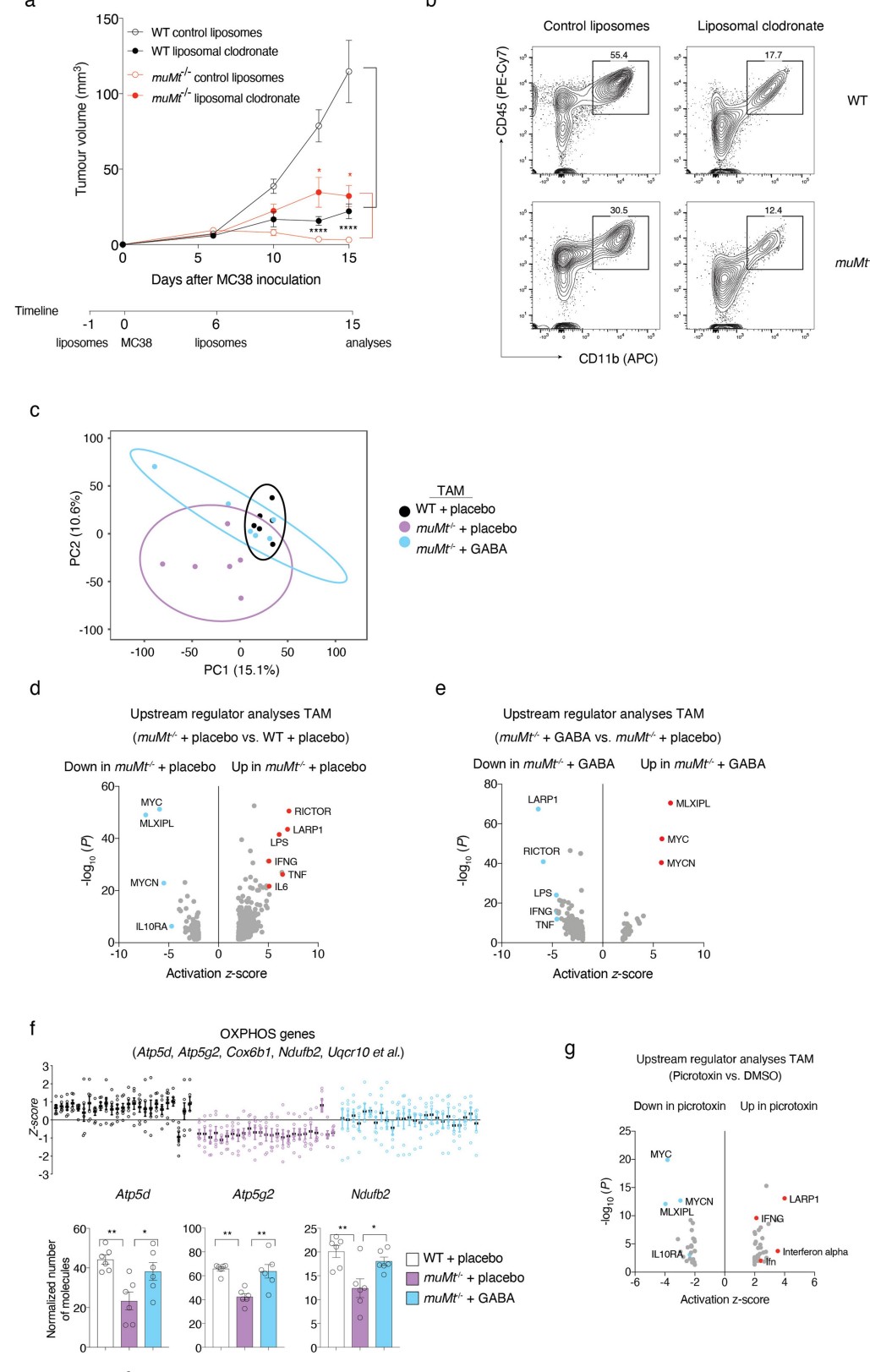

**Extended Data Fig. 5** | See next page for caption.

**Extended Data Fig. 5 | Macrophage involvement in MC38 tumor model.**
**a, b**, Macrophage depletion was performed by injection of liposomal clodronate on day −1 and day 6 after MC38 inoculation in WT or *muMt*[-/-] mice. Tumor volume was monitored at the indicated time points (*n* = 9 (WT control liposomes), 6 (WT liposomal clodronate) or 7) (**a**), flow cytometric analysis was performed on 15 days after tumor inoculation (**b**). **c**, Principal component analysis of transcriptome profile of tumor associated macrophage (TAM) from WT + placebo, *muMt*[-/-] + placebo or *muMt*[-/-] + GABA group of mice at 7 days after MC38 inoculation (*n* = 6). **d, e**, Upstream regulator analysis performed by Ingenuity Pathway Analysis (IPA) and based on the differently expressed genes (DEGs) shown for TAM of *muMt*[-/-] + placebo group compared to WT + placebo group (**d**) or *muMt*[-/-] + GABA group compared to *muMt*[-/-] + placebo group (Wald test *P* < 0.05; two-sided) (**e**), -log$_{10}$ (*P* value; right-tailed Fisher's Exact Test) and activation *z*-score of each predicated upstream regulators are plotted (red, transcripts predicted to be activated; blue, transcripts predicted to be inhibited) (*n* = 6). **f**, Oxidative phosphorylation (OXPHOS) pathway related DEGs between WT and *muMt*[-/-] + placebo group (left), and the transcript level of representative genes (right) (*n* = 6). **g**, Upstream regulator analysis of DEGs in TAM from picrotoxin treatment and their control group 7 days after MC38 tumor inoculation (*n* = 6 from 4 biologically independent animals). \**P* < 0.05, \*\**P* < 0.01, \*\*\*\**P* < 0.0001 (two-way ANOVA (**a**); one-way ANOVA (**f**)). Bars represent mean ±SEM. *n* indicates biological replicates (**a-f**). Data are representative of two experiments (**a**). Exact *P* values are in Source Data.

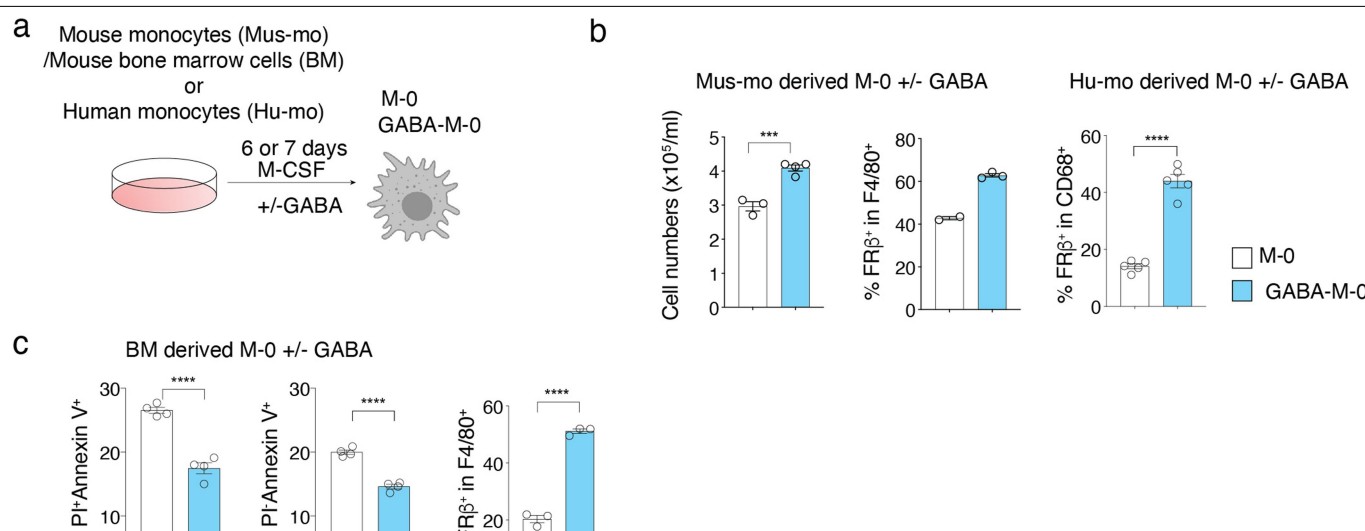

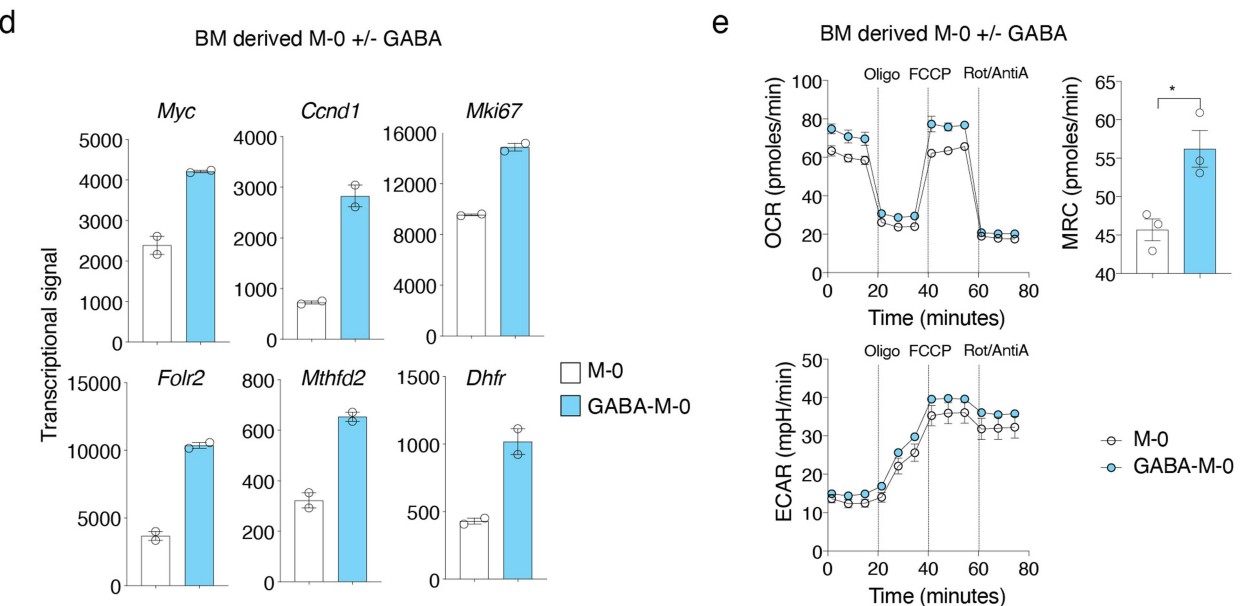

**Extended Data Fig. 6 | GABA increases the proliferation, survival and bioenergetics of macrophages differentiated *in vitro*. a**, Scheme of mouse bone marrow (BM)-derived monocytes (Mus-mo), total BM cell or human blood-derived monocytes (Hu-mo) differentiation in the presence of M-CSF without (M-0) or with GABA (GABA-M-0). **b**, The cell number and the ratio of FRβ⁺ cells in mouse ($n$ = 2 (M-0 (middle), 4 (GABA-M-0 (left) or 3) or human monocytes-derived M-0 ($n$ = 5) with or without GABA treatment. **c**, Quantification of the frequencies of propidium iodide (PI)$^{+or-}$, annexin V⁺ ($n$ = 4) and FRβ⁺ cells ($n$ = 3) by flow cytometry in mouse BM-derived M-0 cells

conditioned with or without GABA. **d**, Transcripts related to cell cycle and folate metabolism obtained from gene chip analysis of M-0 and GABA-M-0 and represented relative to M-0 ($n$ = 2). **e**, Real-time oxygen consumption rate (OCR), maximal respiratory capacity (MRC), and extracellular acidification rate (ECAR) as assessed by Seahorse assay. *$P$ < 0.05, **$P$ < 0.01, ***$P$ < 0.001, ****$P$ < 0.0001 (two-tailed unpaired t-test). Bars represent mean ±SEM. $n$ indicates biological (**d**) or technical replicates. Data are representative of three experiments (**c, e**). Exact $P$ values are in Source Data. The image of immune cell (Extended Figures) and mouse (Fig. 1a) was created with BioRender.com.

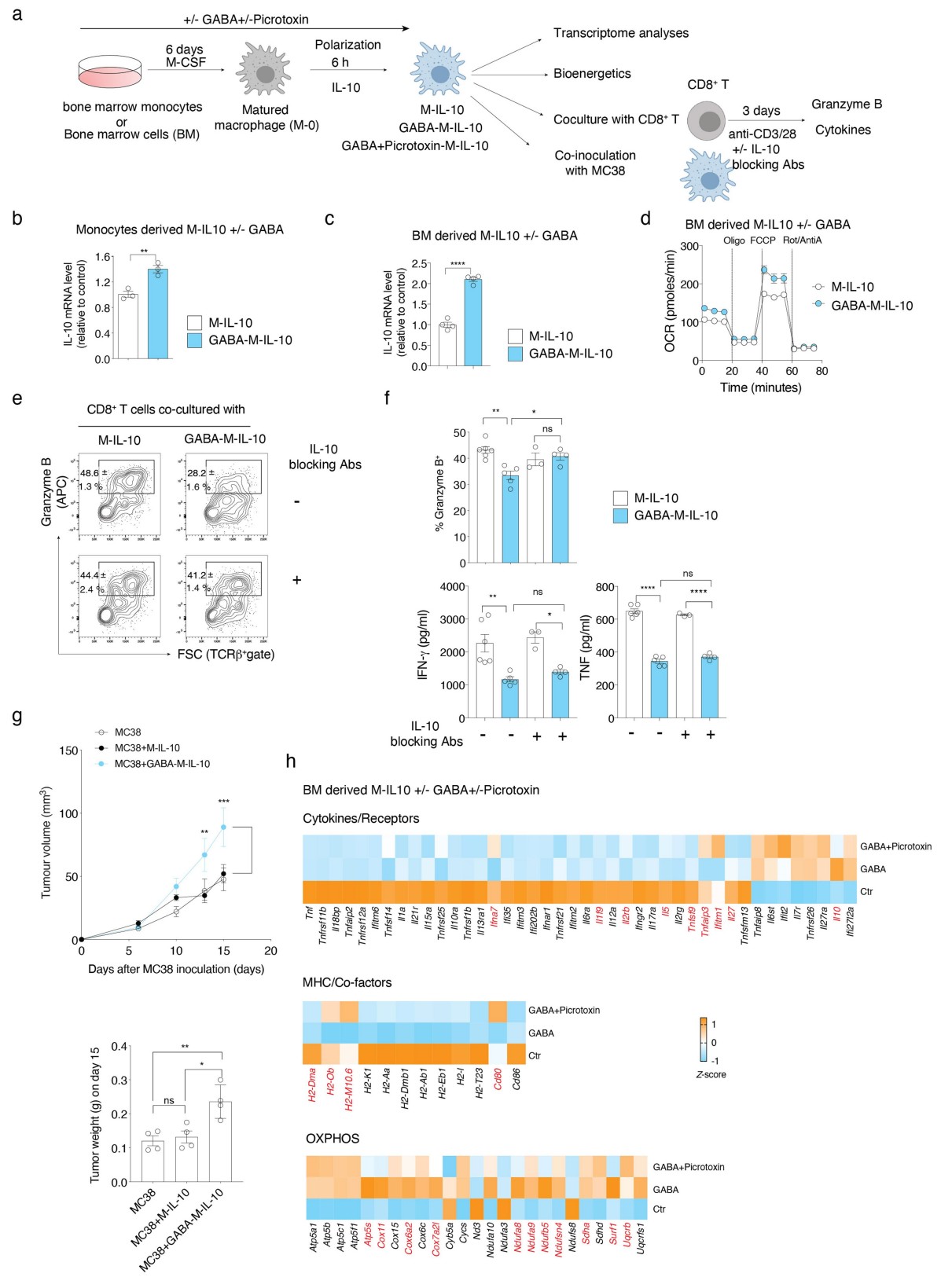

**Extended Data Fig. 7 |** See next page for caption.

**Extended Data Fig. 7 | GABA enhances the anti-inflammatory phenotype of IL-10 polarized macrophages partially via GABA$_A$ receptor signaling.**
**a**, Scheme of bone marrow cell differentiation in the presence of M-CSF without or with GABA for 6 days, followed by 6 h polarization with IL-10 (M-IL-10 and GABA-M-IL-10). qPCR or gene chip analysis was performed for GABA conditioned macrophages with IL-10 polarization (GABA-M-IL-10) and its control group (M-IL-10) ($n = 3$). **b, c**, *Il10* mRNA level measured by q-PCR in monocytes ($n = 3$) (**b**) or BM ($n = 4$) (**c**)-derived M-IL-10 and GABA-M-IL-10. Results are represented relative to control group. **d**, Oxygen consumption rate (OCR) of M-IL-10 and GABA-M-IL-10 measured in real time by extracellular flux analyzer. Oligomycin (Oligo), FCCP and rotenone/antimycin A (Rot/AntiA) were added at the indicated time points ($n = 2$). **e, f**, M-IL-10 and GABA-M-IL-10 conditioned macrophages were co-cultured with CD8$^+$ T cells stimulated for 72 h with anti-CD3 and anti-CD28 in the presence or absence of anti-IL-10 blocking antibodies (Abs). The percentage of granzyme B$^+$ cells in CD8$^+$ T cells analyzed by flow cytometry (**e**) and the concentration of IFNγ and TNF in supernatant measured by Cytometric Bead Array (CBA) (**f**) ($n = 6$ (M-IL-10 + CD8$^+$), 5 (GABA-M-IL-10 + CD8$^+$), 3 (M-IL-10 + CD8$^+$ + IL-10 blocking Abs) or 4 (GABA-M-IL-10 + CD8$^+$ + IL-10 blocking Abs)). **g**, Tumor volume ($n = 6$) or weight ($n = 4$) of WT mice co-injected with MC38 cells together with in vitro generated M-IL-10 or GABA-M-IL10. **h**, The DEGs representing cytokines and their receptors, MHC molecules and co-factors and OXPHOS between GABA and control (Ctr) group, obtained from gene chip analyses ($n = 3$) and shown for all groups as heatmap. Those genes, which are not differentially expressed between GABA + picrotoxin and Ctr group (two-tailed unpaired t-test; $P > 0.05$), are represented as red color (the transcript level is transformed to $z$-score). *$P < 0.05$, **$P < 0.01$, ***$P < 0.001$, ****$P < 0.0001$ (two-tailed unpaired t-test (b, c); one-way ANOVA (**f, g** (right)); two-way ANOVA (**g** (left)). Bars represent mean ±SEM. $n$ indicates biological (**g, h**) or technical replicates (**b-f**). Data are pooled from two experiments (**c**) or representative of two experiments (**d-f**). Exact $P$ values are in Source Data.

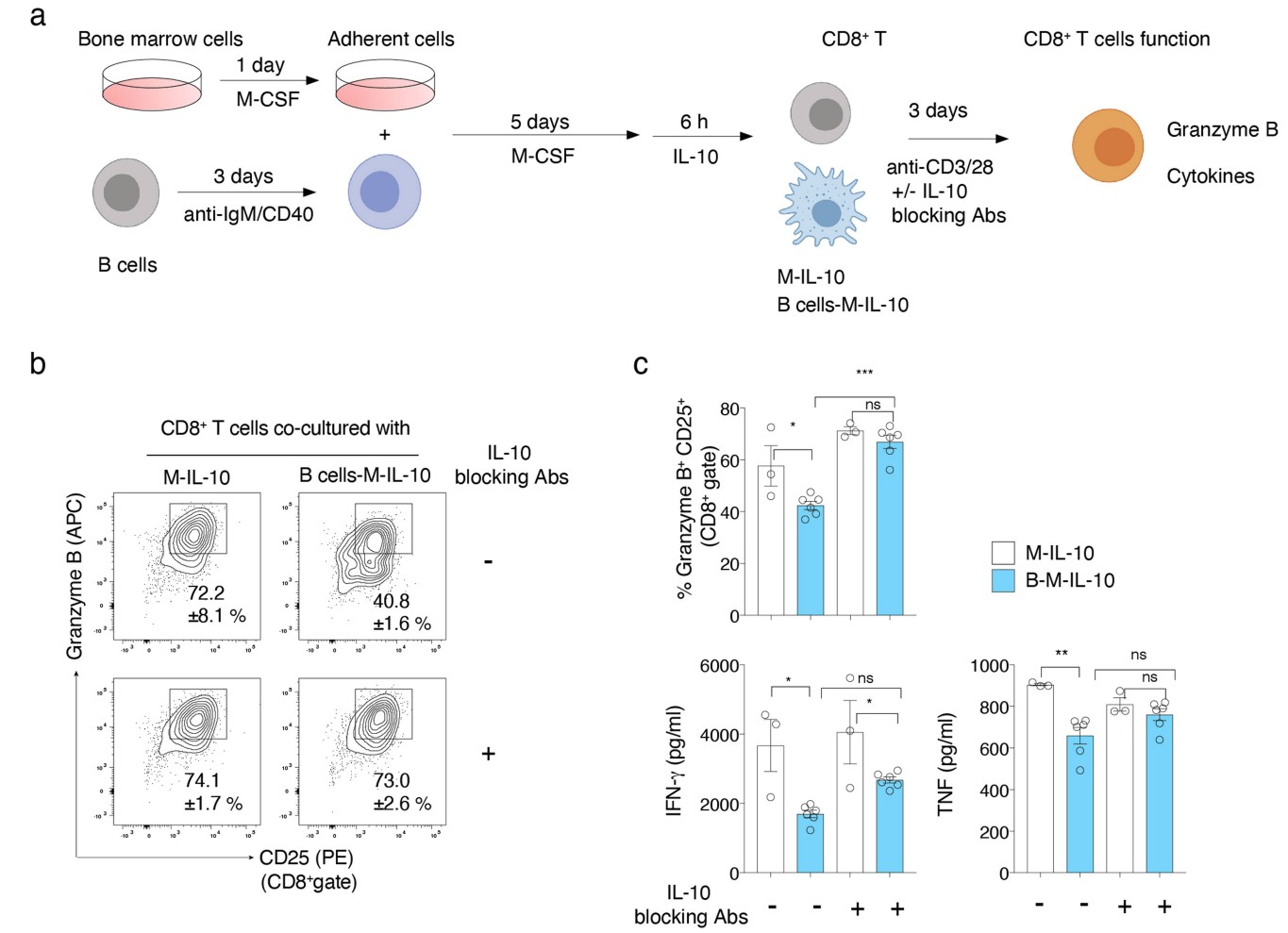

**Extended Data Fig. 8 | Macrophages conditioned by activated B cells suppress the cytotoxic function of CD8+ T cells. a**, Scheme of macrophage conditioning with activated B cells followed by polarization with IL-10 (M-IL-10 and B cells-M-IL-10) and co-culture with CD8+ T cells stimulated with anti-CD3 and anti-CD28 for 72 h in the presence or absence of IL-10 blocking antibodies (Abs). **b, c**, Representative flow cytometry profiles of CD8+ T cells stained for granzyme B and CD25 (**b**), and quantification of the granzyme B+ CD25+ cells in CD8+ gates (upper graph) or the concentration of IFNγ and TNF in the culture supernatants (lower graphs) ($n$ = 3 (M-IL-10 + CD8+ +/- IL-10 blocking Abs) or 6 (B-M-IL-10 + CD8+ +/- IL-10 blocking Abs)) (**c**) are shown. ns: not significant, *$P$ < 0.05, **$P$ < 0.01, ***$P$ < 0.001 (one-way ANOVA). Bars represent mean ±SEM. Circles on graphs indicate an individual sample. $n$ indicates technical replicates. Data are representative of two experiments. Exact $P$ values are in Source Data.

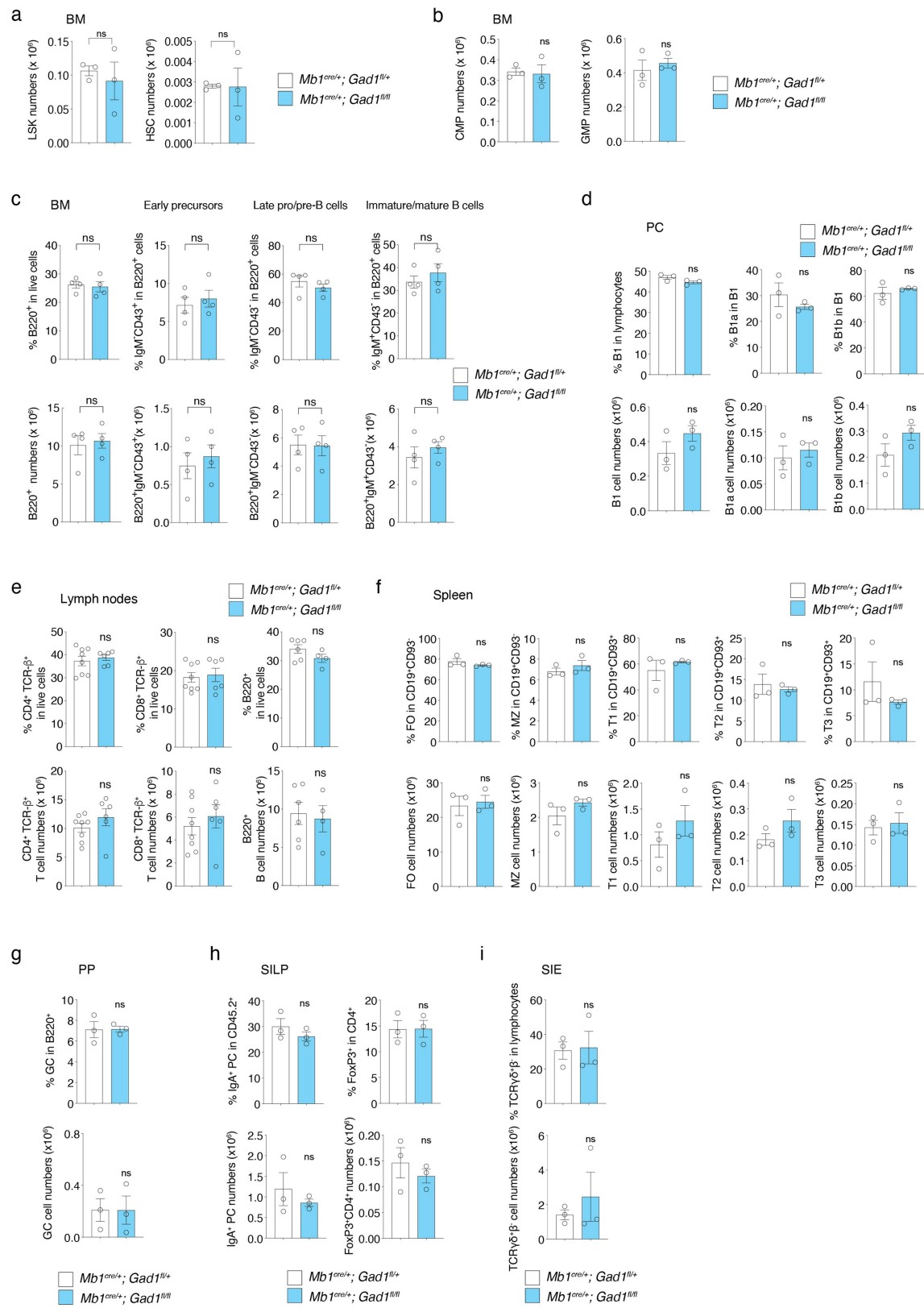

**Extended Data Fig. 9 | Characterization of precursor cells and immune cell subsets in peripheral organs of *Mb1^{cre/+};Gad1^{fl/fl}* mice. a–c**, The frequencies and number of the indicated precursor cells (*n* = 3) (**a, b**) and B cell populations (*n* = 4) (**c**) in bone marrow from *Mb1^{cre/+};Gad1^{fl/+}* or *Mb1^{cre/+};Gad1^{fl/fl}* mice analyzed by flow cytometry. **d–i**, The frequencies and numbers of the indicated immune cell subsets in peritoneal cavity (PC) (*n* = 3) (**d**), lymph nodes (*n* = 8) (**e**), spleen (*n* = 3) (**f**), Peyer's patch (PP) (*n* = 3) (**g**), the small intestine lamina propria (SILP) (*n* = 3) (**h**) and the small intestine epithelial cells (SIE) (*n* = 3) (**i**) from *Mb1^{cre/+};Gad1^{fl/+}* or *Mb1^{cre/+};Gad1^{fl/fl}* mice analyzed by flow cytometry. ns: not significant (two-tailed unpaired t-test). Bars represent mean ±SEM. *n* indicates biological replicates. Data are pooled from two experiments (**c, e**). Exact *P* values are in Source Data.

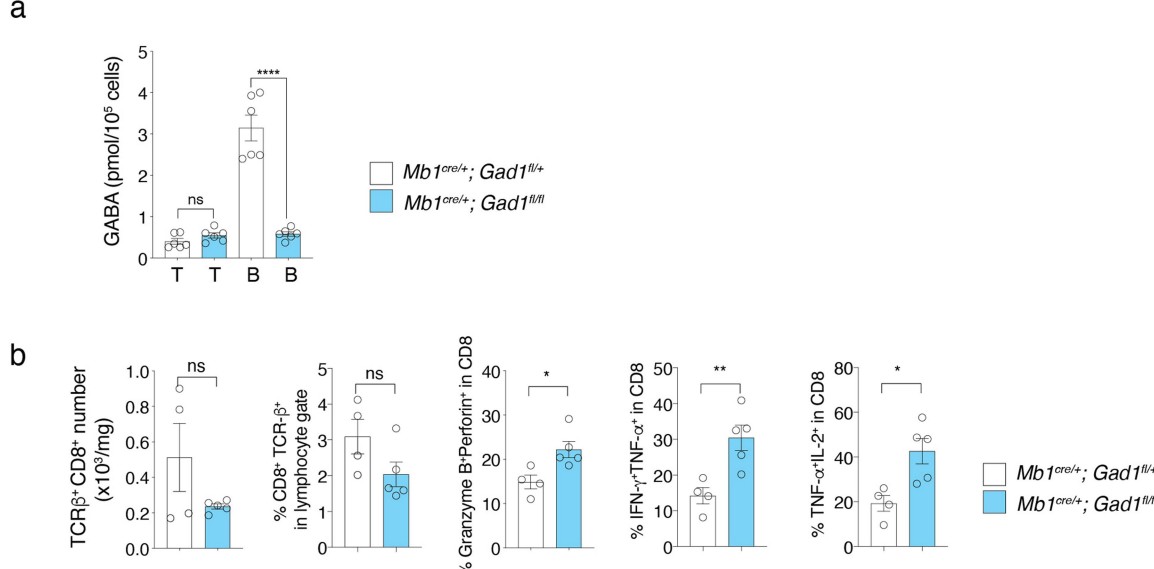

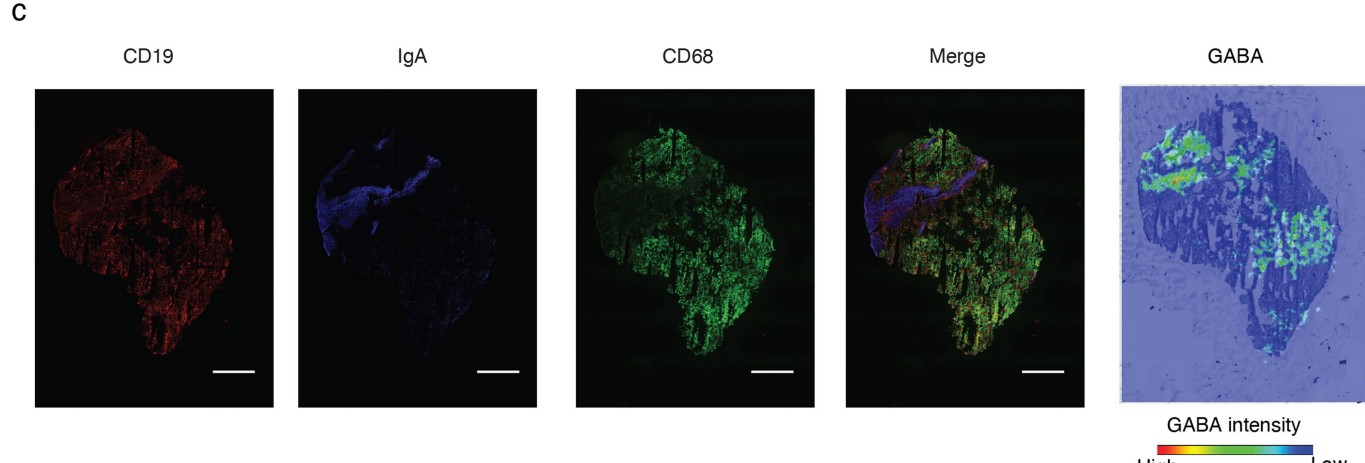

**Extended Data Fig. 10 | Enhanced effector properties of CD8+ T cells infiltrating the tumors in *Mb1^cre/+^;Gad1^fl/fl^* mice and visualization of GABA in human renal cell cancer infiltrated with B cells and IgA+ plasma cells. a**, Intracellular GABA quantitation measured by mass spectrometry in FACS purified T or B cells from the LN of naïve *Mb1^cre/+^;Gad1^fl/+^* or *Mb1^cre/+^;Gad1^fl/fl^* mice (*n* = 6 from 3 biologically independent mice). **b**, The numbers of CD8+ T cells or frequencies of CD8+ T cells in lymphocyte gate and of granzyme B+perforin+, IFNγ+TNFα+ and TNFα+IL-2+ in CD8+ T cell gate cells isolated from tumors of *Mb1^cre/+^;Gad1^fl/+^* (*n* = 4) or *Mb1^cre/+^;Gad1^fl/fl^* (*n* = 5) mice. **c**, Imaging mass spectrometry of GABA and immunohistochemical analyses of human renal cell cancer tissue sections. Scale bar, 1 mm. ns: not significant, *$P$ < 0.05, **$P$ < 0.01, ****$P$ < 0.0001 (two-tailed unpaired t-test). Bars represent mean ±SEM. *n* indicates biological replicates (**b**). The experiment is performed once (**c**). Exact $P$ values are in Source Data.

# Reporting Summary

## Statistics

For all statistical analyses, confirm that the following items are present in the figure legend, table legend, main text, or Methods section.

| n/a | Confirmed | |
|---|---|---|
| ☐ | ☒ | The exact sample size (*n*) for each experimental group/condition, given as a discrete number and unit of measurement |
| ☐ | ☒ | A statement on whether measurements were taken from distinct samples or whether the same sample was measured repeatedly |
| ☐ | ☒ | The statistical test(s) used AND whether they are one- or two-sided<br>*Only common tests should be described solely by name; describe more complex techniques in the Methods section.* |
| ☐ | ☒ | A description of all covariates tested |
| ☐ | ☒ | A description of any assumptions or corrections, such as tests of normality and adjustment for multiple comparisons |
| ☐ | ☒ | A full description of the statistical parameters including central tendency (e.g. means) or other basic estimates (e.g. regression coefficient) AND variation (e.g. standard deviation) or associated estimates of uncertainty (e.g. confidence intervals) |
| ☐ | ☒ | For null hypothesis testing, the test statistic (e.g. *F*, *t*, *r*) with confidence intervals, effect sizes, degrees of freedom and *P* value noted<br>*Give P values as exact values whenever suitable.* |
| ☒ | ☐ | For Bayesian analysis, information on the choice of priors and Markov chain Monte Carlo settings |
| ☐ | ☒ | For hierarchical and complex designs, identification of the appropriate level for tests and full reporting of outcomes |
| ☐ | ☒ | Estimates of effect sizes (e.g. Cohen's *d*, Pearson's *r*), indicating how they were calculated |

*Our web collection on statistics for biologists contains articles on many of the points above.*

## Software and code

Policy information about availability of computer code

| | |
|---|---|
| Data collection | Aria II flow cytometry system was used for collection of FACS data. LC-MS and HPLC (Ultimate3000 system) were used for collection of metabolite data. BZ-X700 fluorescence microscope was used for collection of photomicrograph. LightCycler 96 (SN: 12718) was used for collection q-PCR data. MiSeq System was used for collection RNA sequencing data. Q-Exactive Plus Orbitrap mass spectrometer with a Nanospray Flex ion source coupled to an EASY-nLC 1200 system was used for collection of proteome data |
| Data analysis | Flowjo software (10.7.1) was used for FACS analysis. BZ-X analyzer software was used for analysis of photomicrograph. Compound Discoverer 2.0 software was used for non-target analysis of mass spectrometry data. Proteome Discoverer (2.4) was used for analysis of proteome data. MetaboAnalyst 5.0 was used for analysis of metabolome data. LightCycler 96 SW 1.1 software was used for analysis of q-PCR data. DEseq2(ver. 1.30.1) was used for analysis of RNA sequencing data. Ingenuity Pathway Analysis (01-18-06) was used for the pathway and upstream regulator analysis of RNA sequencing and proteome data. PRISM 8 software was used for statistical analysis. |

For manuscripts utilizing custom algorithms or software that are central to the research but not yet described in published literature, software must be made available to editors and reviewers. We strongly encourage code deposition in a community repository (e.g. GitHub). See the Nature Portfolio guidelines for submitting code & software for further information.

## Data

Policy information about availability of data

All manuscripts must include a data availability statement. This statement should provide the following information, where applicable:

- Accession codes, unique identifiers, or web links for publicly available datasets
- A description of any restrictions on data availability
- For clinical datasets or third party data, please ensure that the statement adheres to our policy

Source data for quantifications represented in all graphs plotted in figures and extended data figures are available in the online version of the paper. The RNA-seq datasets analyzed are publicly available in the Gene Expression Omnibus repository with the accession numbers GSE169543 and GSE183246 (released on September 02, 2021). The Gene chip datasets are provided in Supplementary Table. The proteomics datasets are available via ProteomeXchange with identifier PXD028403 (released on the date of online publication). The DESeq2 (1.30.1) package was used for analyzing RNA-seq data (https://bioconductor.org/packages/release/bioc/html/DESeq2.html).

# Field-specific reporting

Please select the one below that is the best fit for your research. If you are not sure, read the appropriate sections before making your selection.

☒ Life sciences          ☐ Behavioural & social sciences          ☐ Ecological, evolutionary & environmental sciences

For a reference copy of the document with all sections, see nature.com/documents/nr-reporting-summary-flat.pdf

# Life sciences study design

All studies must disclose on these points even when the disclosure is negative.

| | |
|---|---|
| Sample size | No sample size calculation was performed. The number of animals was determined based on the number of animals implemented in previously published papers. The number of human samples was determined based on availability. Each experiment was replicated for subsequent statistical analysis. |
| Data exclusions | Infrequently, mice showed signs of inflammation even in normal SPF condition. Therefore, when sacrificed, mice were routinely checked for the inflammation status and samples from mice showing severe inflammation status (splenomegaly and colitis) were excluded. |
| Replication | Each animal experiment was performed with at least 3 biological replicates. Clinical human experiment was performed with at least 3 biological replicates except for imaging MS. All attempts at replication gave similar results and reliably reproduced. |
| Randomization | Age and sex matched mice were randomly allocated into experimental groups. In human study, patients with RA were randomly recruited after informed consent and selected based on the availability of plasma sample, then patients together with symptoms other than RA (stroke, herpes zoster, dementia, cancer, hemodialysis, pneumonia, surgery in a year) and under steroid treatment were removed from the analysis to investigate pure RA effect on metabolites. Since ratio of male and female is 1:5.8 in the subject group, we focused on female to avoid sex bias. |
| Blinding | Blinding was performed in the measurement of the tumor size.<br>Otherwise blinding was not used since the data collection and the analysis were performed with quantitative instruments to maintaining objectivity. |

# Reporting for specific materials, systems and methods

We require information from authors about some types of materials, experimental systems and methods used in many studies. Here, indicate whether each material, system or method listed is relevant to your study. If you are not sure if a list item applies to your research, read the appropriate section before selecting a response.

## Materials & experimental systems

| n/a | Involved in the study |
|---|---|
| ☐ | ☒ Antibodies |
| ☐ | ☒ Eukaryotic cell lines |
| ☒ | ☐ Palaeontology and archaeology |
| ☐ | ☒ Animals and other organisms |
| ☐ | ☒ Human research participants |
| ☐ | ☒ Clinical data |
| ☒ | ☐ Dual use research of concern |

## Methods

| n/a | Involved in the study |
|---|---|
| ☒ | ☐ ChIP-seq |
| ☐ | ☒ Flow cytometry |
| ☒ | ☐ MRI-based neuroimaging |

# Antibodies

| Antibodies used | (Flow Cytometry)<br>APC-Cy7-anti-CD8a (Biolegend, clone 53-6.7, # 100713, 1:100), APC-anti-TCR-β (Biolegend, clone H57-597, # 109211, 1:50 ), Brilliant Violet 570-anti-CD4 (Biolegend, clone RM4-5, # 100542, 1:100), Alexa Fluor 700-anti-CD62L (Biolegend, clone MEL-14, # 104426, 1:100), APC-nti-CD11c (Biolegend, clone N418, 117309, 1:100), APC-anti-CD11b (Biolegend, clone M1/70, # 101211, 1:100), anti-CD3ε (Biolegend, clone 145-2C11), PE-Cy7-anti-CD45.2 (Biolegend, clone 104, #109829, 1:100), APC-anti-Granzyme B (Biolegend, clone QA16A02, # 372203, 1:100), PE-anti-Perforin (Biolegend, clone S16009B, # 154405, 1:100), Alexa Fluor 488-anti-F4/80 (Biolegend, clone BM8, # 123119, 1:100), FITC-anti-cKit (Biolegend, clone 2B8, #105805, 1:100), PE-Cy7-anti-SCA-1 (Biolegend, clone D7, #108113, 1:100), PE-anti-CD48 (Biolegend, clone HM48-1, # 103405, 1:100), Pacific Blue-anti-CD150 (Biolegend, clone TC15-12F12.2, # 115923, 1:100), PE-Cy7-anti-CD93 (Biolegend, clone AA4.1, #136505, 1:200), PE-Cy7-anti-CD38 (Biolegend, clone 90, # 102718, 1:100), APC-anti-IFN-γ (eBioscience, clone XMG1.2, #16-7311, 1:100), PE-anti-CD44 (eBioscience, clone IM7, #12-0441, 1:100), APC-anti-B220 (eBioscience, clone RA3-6B2, # 17-0452, 1:100), eFluor450-anti-B220 (eBioscience, clone RA3-6B2, # 48-0452,1:100), Brilliant Violet 570-anti-B220 (Biolegend, clone RA3-6B2, #103237, 1:100), Alexa Fluor 488-anti-B220 (eBioscience, clone RA3-6B2, # 53-0452, 1:100), PerCP eFluor710-anti-TNF-α (eBioscience, clone MP6-XT22, # 46-7321, 1:100), eFluor450-anti-IgD (eBioscience, clone 11-26c, #48-5993, 1:200), eFluor 450-anti-CD21/CD35 (eBioscience, clone eBio4E3, # 48-0212, 1:200), anti-FOXP3 (eBioscience, clone FJK-16s), PE-anti-CD25 (BD Biosciences, clone PC61, # 553866, 1:50), PE-anti-IL-2 (BD Biosciences, clone JES6-5H4, # 554428, 1:100), PE-anti-CD23 (BD Biosciences, clone B3B4, # 553139, 1:200), PE-Cy7-anti-CD43 (BD Biosciences, clone S7, # 01605B, 1:100, FITC-anti-CD16/32 (BD Biosciences, clone 2.4G2, # 553144, 1:100 ), APC-anti-CD19 (BD Biosciences, clone 1D3, #550992, 1:100), anti-CD5 (BD Biosciences, clone 53-7.3), anti-CD95 (BD Biosciences, clone Jo2), anti- γδ TCR (BD Biosciences, clone GL3), FITC-anti-IgM (SouthernBiotech, polyclonal, #1022-02, 1:100) and PE-anti-IgA (SouthernBiotech, polyclonal, # 1040-09, 1:200).<br><br>(Sorting)<br>biotin-anti-CD20 (Biolegend, clone 2H7, #302349, 1:100) , biotin-anti-CD19 (Biolegend, clone HIB19, # 302203, 1:100)<br><br>(Culture) concentration as described in Method<br>anti-CD3 (BD Biosciences, 145-2C11, #567115)<br>anti-CD28 (BD Biosciences, 37.51, # 553294)<br>anti-IgM (Jackson ImmunoReserch,)<br>anti-CD40 (BD Biosciences,3/23 or HM40-3, # 553787 or 553721)<br>F(ab')2-Goat anti-human IgG/IgM (Invitrogen,)<br>anti-IL-10 blocking antibodies (eBioscience, JES5-2A5, # 16-7102)<br>(Imaging of NFkB)<br>anti-human total p65 rabbit antibody (Cell Signaling Technology, clone D14E12, #8242, 1:200)<br>Alexa Fluor 594-conjugated F(ab')2 fragments of goat anti-rabbit IgG (H+L) (Thermo Fisher Scientific, #1:1000)<br><br>(Immunofluorescence)<br>anti-mouse CD3ε (BD Pharmingen, clone 500A2, #553239, 1:25), anti-mouse B220 (eBioscience, clone RA3-6B2, # 13-0452, 1: 50), anti-mouse CD11c (eBioscience, clone N418, # 13-0114-85, 1: 50),<br>anti-human CD68 (eBioscience, clone 815CU17, #13-0687, 1: 50),  anti-human CD19 (abcam, clone EPR5906, #ab134114, 1:100), anti-human IgA (SouthernBiotech, polyclonal, # 2052-31, 1: 100) |
|---|---|
| Validation | All antibodies from commercial vendors were validated by the manufacturers on their websites. |

# Eukaryotic cell lines

Policy information about cell lines

| Cell line source(s) | MC38 (murine colon adenocarcinoma) cell line was provided by James P. Allison of the Memorial Sloan Kettering Cancer Center.<br>Jurkat, clone E6-1 was obtained from Art Weiss(UCSF, CA, USA). |
|---|---|
| Authentication | The cell lines were not authenticated |
| Mycoplasma contamination | MC38 cell lines were tested, and they were mycoplasma free. Jurkat cell lines were not tested. |
| Commonly misidentified lines<br>(See ICLAC register) | No commonly misidentified cell lines were used. |

# Animals and other organisms

Policy information about studies involving animals; ARRIVE guidelines recommended for reporting animal research

| Laboratory animals | WT (C57BL/6J or C57BL/6N), muMt-/- mice (C57BL/6J), Cd3e-/- mice (C57BL/6J), rag1-/-mice (C57BL/6J), mb1cre/+- gad1fl/+ and mb1cre/+- gad1fl/fl mice were used in this study. Both male and female mice were used. 2 Mo-5 Mo old mice were used in this study. The SPF facility of RIKEN is maintained in a 12-hour light, 12-hour dark cycle at 23 ± 2 °C with 50 ± 10% humidity. |
|---|---|
| Wild animals | The study did not involve wild animals. |
| Field-collected samples | The study did not involve samples collected from the field. |

| Ethics oversight | All experiments were conducted in accordance with protocols approved by the Institutional Animal Care and Use Committee of the RIKEN Yokohama Branch. |

Note that full information on the approval of the study protocol must also be provided in the manuscript.

## Human research participants

Policy information about studies involving human research participants

| Population characteristics | For blood cell study, healthy volunteers of male and female and age 20-40 with various genetic background were participated. For RA study, Japanese female diagnosed as RA from age 31-75 were participated. |
| Recruitment | The students and researchers in Kyoto university were recruited as the healthy volunteers through the oral announcement of the study. Since the participants were age 20-40, it may affect the interpretation of the results in terms of generalization to a broader age. Rheumatoid arthritis patients were enrolled in the Kyoto University Rheumatoid Arthritis Management Alliance (KURAMA) cohort and all were Japanese. The genetic background may affect the interpretation of the result. |
| Ethics oversight | All experiments were conducted in accordance with protocols approved by ethical committee of RIKEN. |

Note that full information on the approval of the study protocol must also be provided in the manuscript.

## Clinical data

Policy information about clinical studies

All manuscripts should comply with the ICMJE guidelines for publication of clinical research and a completed CONSORT checklist must be included with all submissions.

| Clinical trial registration | Kyoto University Rheumatoid Arthritis Management Alliance (KURAMA) cohort |
| Study protocol | http://allie.dbcls.jp/pair/KURAMA;Kyoto+University+Rheumatoid+Arthritis+Management+Alliance.html |
| Data collection | The description was provided in the manuscript. |
| Outcomes | The description was provided in the manuscript. |

## Flow Cytometry

### Plots

Confirm that:

☒ The axis labels state the marker and fluorochrome used (e.g. CD4-FITC).

☒ The axis scales are clearly visible. Include numbers along axes only for bottom left plot of group (a 'group' is an analysis of identical markers).

☒ All plots are contour plots with outliers or pseudocolor plots.

☒ A numerical value for number of cells or percentage (with statistics) is provided.

### Methodology

| Sample preparation | Mice cells were prepared from spleen, lymph node(LN), small intestine(SI) and bone marrow(BM). For making single cell suspension, spleen and LN were mashed, BM cells were flushed out using syringe and needles from femurs and tibiae, and SI was digested with collagenase (1.5 mg/ml; 30 min twice). Tumor tissues were minced and digested with collagenase (1.5 mg/ml; 30 min once). |
| Instrument | BD Aria II flow cytometry system was used for collection of FACS data |
| Software | Flowjo software (10.7.1) was used for FACS analysis. |
| Cell population abundance | More than 200,000 cells of the targeted populations were sorted using the high purity mode. |
| Gating strategy | Using the FSC/SSC gating, debris was removed, and the single alive cells were gated. Each population was gated based on the surface or intracellular markers as described in the manuscript. |

☒ Tick this box to confirm that a figure exemplifying the gating strategy is provided in the Supplementary Information.

