## [Peer Review File · Nature]

Manuscript Title: B cell-derived GABA elicits IL-10+ macrophages limiting anti-tumor immunity

Redactions – unpublished data

Reviewer Comments & Author Rebuttals

Reviewer Reports on the Initial Version:

Referee #1 (Remarks to the Author):

The authors begin with a clever basic experiment in which they performed a classic mouse foot-pad immunization and then brought to bear state-of-the-art technologies to compare the metabolites in the ipsilateral lymph node (LN) near to the immunization site versus those in the contralateral resting LN. They found 200 metabolites that differed significantly between these LNs, and a stand-out difference was activation of the glutamate pathway in the ipsilateral LN. Next, they assessed the contribution of different immune cell subsets to these metabolite differences by repeating the immunization studies with mice lacking T cells, B cells, or all mature B and T cells. Interestingly, the LN of B cell-deficient mice had the most distinct metabolic changes, and surprisingly, GABA, which heretofore was not known to be synthesized by B cells, was the major metabolite that was upregulated in the ipsilateral LN in a B cell-dependent manner. These results suggest that antigenic stimulation induces B cell synthesis of GABA. Using labeled glutamine, the authors traced its catabolism in activated B cells and found that glutamine was converted to glutamate (the substrate of the GABA synthetic enzyme glutamic acid decarboxylase (GAD65/67)) and that labeled GABA was almost exclusively detected in B cells, but not T cells. These data suggest that antigenic stimulation increases GAD enzymatic activity in B cells leading to greater production of the chemical transmitter GABA.

They then examined GABA's effects on anti-tumor responses using a common cancer model in which B cells are known to inhibit anti-tumor responses. As expected, B-cell deficient mice controlled tumor growth better than wild-type mice, and histological analysis showed expansion of tumor-infiltrating CD8+ T cells with enhanced cytotoxic and inflammatory markers. Implantation of a slow GABA-releasing pellet led to an increase in tumor size in B-cell deficient mice compared to controls, which was accompanied by reduced frequency of CD8+ T cells and their production of cytotoxic and inflammatory markers.

Importantly, they also observed that tumor-associated macrophages (TAMs) in B-cell deficient mice had enhanced expression of the TNF-signaling pathway compared to wild-type mice, and that this phenotype was dampened when B cell-deficient mice received GABA. The findings that the modulation of TNF pathways is B cell-dependent and related to changes in mitochondrial respiration are novel and interesting.

Finally, taking advantage of mice with conditional deficiencies in B cell GABA production (GAD67^{-/-} mice) they show that 1) their B cells have reduced intracellular GABA levels, 2) tumor cells implanted into these mice have significantly reduced growth, and 3) their tumor-infiltrating CD8+ T cells had enhanced cytotoxic and pro-inflammatory properties. Together, these results indicate that B-cell-produced GABA significantly contributes to limiting anti-tumor responses. The use of state-of-the-art technologies in this report is impressive and led to new observations that have previously gone unnoticed. Their major observation, that B-cell-produced GABA significantly contributes to limiting anti-tumor responses is a novel and unexpected finding. It points to new approaches to increase the efficacies of immunotherapies for solid tumors. According, the report will be of wide interest to the Nature readership at a basic science level as well as for its potential clinical applications.

Major concern:

The authors rule out a direct effect of GABA on CD8+ T cells based on a study shown in Extended

data Fig. 6. There is, however, long-standing evidence that CD8+ T cell functions are directly modulated by activation of their GABAA-Rs. First, GABA was found to modulate CTL activity [1]. Second, electrophysiological studies by Birnir and colleagues [2] observed that physiological levels of GABA activated GABAA-R channels on CD8+ T cells at a single-cell level using patch-clamping. These GABA-Rs displayed channel conductance that was typical of extra-synaptic GABAA-Rs. GABA's effects were enhanced by a GABAA-R positive allosteric modulator further confirming the effects were GABAA-R mediated. These results, obtained by a leading GABA-R electrophysiology lab, provide strong functional evidence of a direct action of GABA on CD8+ T cells. Additionally, these authors found that low levels of GABA inhibited CD8+ T cell proliferation, similar to reports from many labs that GABA modulates the proliferation and cytokine/chemokine secretion of CD4+ T cells. Third, single-cell transcriptome data suggests that CD8+ T cells express GABA-Rs since their GABA-R subunit expression is essentially the same as that of CD4+ T cells (e.g., [3] and ImmGen data banks). Finally, GABAA-R agonist administration induces CD8+ Tregs in vivo [4, 5], although that effect could be indirect. Hence, GABA can act directly on CD8+ T cells and this notion should be carried throughout the manuscript when interpreting their observations. The author's findings shown in extended data Fig 6 indicate that GABA treatment actually tended to reduce CD8+ T cell secretion of IFN γ and TNF in vitro. If the authors wish to provide contrary evidence to reports of GABA directly modulating CD8+ T cells, they should further support their negative observations by testing a dose range of GABA (100 nM-1 mM, n=6-7 samples/group, in duplicate or triplicate) in their cytokine and proliferation assays.

Moderate concerns:

A) It is standard practice when studying GABA-Rs to include controls with GABAA-R and GABAB-R specific agonists and antagonists to verify the results are due to effects on GABA-Rs and not off-target effects, and to determine whether observations are GABAA-R or GABAB-R mediated. It would enhance the manuscript, but it is not essential, that at least one of their novel in vitro findings was validated using the GABAA-R agonist muscimol, the GABAB-R agonist baclofen, along with GABA combined with bicuculline or saclofin (GABAA-R and GABAB-R antagonists, respectively).

B). The author's granular studies provide additional depths of knowledge on the actions of GABA on immune cells. For instance, they performed differential gene expression analysis that showed that GABA modulates macrophages toward anti-inflammatory responses. Please discuss these new findings in the context of past reports showing that GABA modulates APC toward anti-inflammatory responses. For instance, studies in the autoimmune and parasitology fields have shown that rodent and human APC express both GABAA-Rs and GABAB-Rs and both GABAA-R [4-10] and GABAB-R [3, 11-14] -specific agonists inhibit their inflammatory activities. Along this line, it is likely that APC in the tumor microenvironment secrete GABA, e.g., their reference 10 showed that macrophages and DC express GAD65 and secrete GABA. The authors should incorporate these findings into their interpretations and discussion.

C) GABA also inhibits Th1 and Th17 cells, while also promoting CD4+ and CD8+ Tregs ([15, 16] and references therein). Notably, administration of GABA-R agonists promotes the spreading of IL-10-secreting autoreactive T cell responses, while inhibiting the spreading of inflammatory T cell autoreactivities, to autoimmune target tissue antigens [15, 17]—these same actions are also likely to impede the spreading of anti-tumor effector responses among tumor antigen determinants. All of these effects on CD4+ T cells can directly attenuate CD4+ T cell-mediated tumor cell killing, as well as reduce help for CD8+ effector T cell responses, and should be made part of their interpretation and discussion.

D) Please check the level of GABA in MC38 cells. Positive controls could include mouse brain cells and activated B cells, and negative controls could include HEK 293, HELA, or COS cells.

Minor issues:

The text makes it seem that there are stark differences in GABA levels between B cells and T cells. However, their data (Extended data Fig. 3A) shows that GABA levels in T cells are only about 40% lower than that of B cells. If B cell GABA has such an immunological impact, it begs the question what is T cell GABA used for? I understand that the authors observed that essentially no labeled glutamine ends up as GABA in activated T cells. However, GABA can be also made from putrescine ([3] and references therein) and can be taken up from the extracellular environment by GABA transporters. Their strong statements that T cells do not make or use GABA do not seem warranted by the information on hand.

Please inform this reviewer why they chose to knock out GAD67 and not GAD65 or both? Their reference 10 showed that mouse APC make GAD65 and GABA. But GAD67 is of interest since it releases GABA in non-vesicular fashion. In any case, they should bear in mind that cytoplasmic GABA can provide ATP via the GABA shunt especially in oxygen-poor conditions, and so GAD67-deficiency doesn't just affect GABA secretion.

The authors make relative comparisons. For example, they say that GABA levels in GAD67^{-/-} B cells were reduced to the "minimal levels observed in T cells". As noted above, the GABA in T cells was only about 40% less than B cells. Relative levels are not useful since GABAergic neurons or macrophages could also be used as reference points. Please discuss the levels of GABA in absolute units.

They note that GABA treatment had no effect on tumor size in wild-type mice and they speculate that B cell-derived GABA production saturates the system such that exogenous GABA does not impede anti-tumor responses. Please note that GABA has been found to inhibit, or conversely, promote the growth of a few different solid tumors, and some mechanistic insights have been obtained (reviewed in [18]). These findings suggest that the effect of GABA on tumors is dependent on the cancer cell phenotype and its intracellular signaling pathways, and not B cells. Along that line, I know of no clinical evidence of commonly prescribed GABA-R modulators modulating cancer incidence or progression.

Line 83 states GABA was the most upregulated metabolite in ipsilateral vs. contralateral. Yet, line 105 states GABA was found at elevated levels in B cells from the contralateral side. Please clarify.

Reviewed by Daniel Kaufman

1. Bergeret, M. et al. (1998) GABA modulates cytotoxicity of immunocompetent cells expressing GABAA receptor subunits. *Biomed Pharmacother* 52 (5), 214-9.
2. Mendu, S.K. et al. (2011) Increased GABA(A) channel subunits expression in CD8(+) but not in CD4(+) T cells in BB rats developing diabetes compared to their congenic littermates. *Mol Immunol* 48 (4), 399-407.
3. Tian, J. et al. (2021) GABAB-Receptor Agonist-Based Immunotherapy for Type 1 Diabetes in NOD Mice. *Biomedicines* 9 (1).
4. Tian, J. et al. (2019) Homotaurine treatment enhances CD4+ and CD8+ Treg responses and synergizes with low-dose anti-CD3 to enhance diabetes remission in type 1 diabetic mice. *ImmuoHorizons*, Oct 21;3(10):498-510.
5. Tian, J. et al. (2018) Homotaurine, a safe blood-brain barrier permeable GABAA-R-specific agonist, ameliorates disease in mouse models of multiple sclerosis. *Sci Rep* 8 (1), 16555.
6. Tian, J. et al. (2011) Oral GABA treatment downregulates inflammatory responses in a mouse model of rheumatoid arthritis. *Autoimmunity* 44, 465-470.
7. Bhat, R. et al. (2010) Inhibitory role for GABA in autoimmune inflammation. *Proc Natl Acad Sci U S A* 107 (6), 2580-5.
8. Januzi, L. et al. (2018) Autocrine GABA signaling distinctively regulates phenotypic activation of mouse pulmonary macrophages. *Cell Immunol* 332, 7-23.
9. Prud'homme, G.J. et al. (2013) GABA protects human islet cells against the deleterious effects

- of immunosuppressive drugs and exerts immunoinhibitory effects alone. *Transplantation* 96 (7), 616-23.
10. Bhandage, A.K. et al. (2020) A motogenic GABAergic system of mononuclear phagocytes facilitates dissemination of coccidian parasites. *Elife* 9.
 11. Huang, S. et al. (2015) The anti-spasticity drug baclofen alleviates collagen-induced arthritis and regulates dendritic cells. *J Cell Physiol* 230 (7), 1438-47.
 12. Duthey, B. et al. (2010) Anti-inflammatory effects of the GABA(B) receptor agonist baclofen in allergic contact dermatitis. *Exp Dermatol* 19 (7), 661-6.
 13. Beales, P.E. et al. (1995) Baclofen, a gamma-aminobutyric acid-b receptor agonist, delays diabetes onset in the non-obese diabetic mouse. *Acta Diabetol* 32 (1), 53-6.
 14. Crowley, T. et al. (2015) Modulation of TLR3/TLR4 inflammatory signaling by the GABAB receptor agonist baclofen in glia and immune cells: relevance to therapeutic effects in multiple sclerosis. *Front Cell Neurosci* 9, 284.
 15. Tian, J. et al. (2021) Homotaurine limits the spreading of T cell autoreactivity within the CNS and ameliorates disease in a model of multiple sclerosis. *Sci Rep* 11 (1), 5402.
 16. Tian, J. et al. (2020) GABA administration prevents severe illness and death following coronavirus infection in mice. *bioRxiv* DOI 10.1101/2020.10.04.325423.
 17. Tian, J. et al. (2014) Combined therapy with GABA and proinsulin/alum acts synergistically to restore long-term normoglycemia by modulating T-cell autoimmunity and promoting beta-cell replication in newly diabetic NOD mice. *Diabetes* 63 (9), 3128-34.
 18. Jiang, S.H. et al. (2020) Neurotransmitters: emerging targets in cancer. *Oncogene* 39 (3), 503-515.

Referee #2 (Remarks to the Author):

Zhang et al. B cell derived GABA elicits anti-inflammatory macrophages and limits anti-tumor cytotoxic responses

This paper starts out with an unbiased look at what metabolites are made in reactive versus non-reactive mouse lymph nodes. Among a variety of metabolites, GABA was prominently made primarily by B cells and less by T cells or myeloid cells. B cell activation seemed critical for GABA expression by B cells. In the MC38 tumor model, tumor growth was reduced in uMT mice and was partially restored by adding exogenous GABA. CD8 T cell function (GZB, PRF, IFN γ and TNF) was increased in uMT mice, and exogenous GABA returned T cell activation back to WT levels. Tumor associated macrophages had very different transcriptomes in WT and uMT mice, but it was less clear whether exogenous GABA restored the B cell phenotype. Nevertheless, genes like TNFRSF1b, IFN γ R, IL-1b were clearly regulated in a GABA-dependent fashion. BM-derived macrophages cultured with GABA had higher folate R expression, increased OXPHOS, less inflammatory cytokines and higher IL-10 and reduced MHC expression. IL-10 was a prominent factor in reducing T cell activation by macrophages. Finally, mice lacking GAD1 specifically in B cells exhibited delayed tumor growth and enhanced T cell function.

The presence or absence of B cells has a much profound effect on tumor growth and macrophage function than can be accounted for by GABA (Fig 3A and D). In particular, the gene expression profiles of TAMs from uMT do not look like they were restored to WT with the addition of GABA (Fig 3D). Nevertheless, the TNF target genes and the OXPHOS genes are at least partly regulated by GABA. Is this a very specific convergence of GABA signaling on TNF signaling pathway?

In Fig 3B, the data is expressed as % of T cells in the CD45+ population. Given that the B cells are missing, the CD45 population is not the appropriate denominator in the uMT mice. This data needs to be expressed as actual number of T cells / tumor or number of CD8 T cells/ tumor.

In Fig 3C, the FACS plots need to contain the % of cells + or - SD in each quadrant or gate.

The graphs do not make sense – is this a % or is the data normalized to control. It looks like the data is normalized to 1 in WT. This should be represented as % of the CD8+ T cell population

Fig 4H needs a statistical analysis – AUC?

Fig 4i The gated areas need to be labeled with percentage + and – SD. Stats?

Is the real functional impact of B cell-derived myeloid cells in the lymph node or in the tumor? The data is all about the tumor, but it may be that the T cells and the myeloid cells are messed up in the lymph node and never make it to the tumor. Maybe the T cells are excluded from the tumor.

New data from Shao 2021 Blood PMID: 32881992 show that GABA acts on B cell development and that GABA agonists enhance hematopoiesis. Are myeloid macrophages and B cells normal in GABA-supplemented mice?

Referee #3 (Remarks to the Author):

This is a very interesting study that reveals anti-inflammatory and pro-tumor effects of B cell derived GABA in mice. GABA is produced at high levels by plasma cells in the lamina propria of the gut, and to a lesser extent by all B cell subsets. GABA counteracts the anti-tumor effect of B cell-deficiency, at least in part by indirectly promoting formation of cytotoxic T cells. Using bone marrow-derived macrophages, the authors show that GABA leads to the formation of anti-inflammatory IL-10-secreting macrophages, that can dampen CD8 T cell cytotoxicity in an IL-10-dependent manner. In mice lacking Gad1 in B cells (that cannot form GABA), tumor formation is greatly reduced. The results are exciting and open up new avenues of research. The conclusions could be strengthened by addressing the issues listed below.

Issues to address.

While most publicly available databases reveal Gad1 RNA expression in mouse B cells, this is not apparent in human B cells. Have the authors looked for GABA production by human B cells? If this pathway is not present in humans, this should be mentioned.

The authors draw the conclusions on direct effect on macrophages from cultures in which total bone marrow cells are incubated with GABA and M-CSF from day 0. These experiments do not exclude the possibility that GABA is acting on other bone marrow cells to influence macrophage differentiation. To conclude GABA acts directly on macrophages, purified primary macrophages obtained from spleen or peritoneal cavity could be used.

In line with the above, to show that the changes in tumor infiltrating CTLs are a consequence of GABA acting on macrophages rather than on the tumor itself, or on other haemopoietic or endothelial cells, tumor growth could be evaluated in mb1.Cre-gad1.fl/fl mice after macrophage depletion.

Have the authors evaluated the effects of GABA on CD4+ Tregs? Dampening their numbers or function in tumors of MuMT mice may be an alternative or additional explanation for promoting tumor growth.

Have the authors looked for expression of PD-L1 on tumor cells (not just macrophages) before and after GABA injection in the MuMT mice to exclude direct actions on the tumor itself?

Only the bone marrow B cell compartment has been characterized in depth in Mb1-Cre.Gad1 fl/fl

mice. It would be important to show normal peripheral B cell compartments in these mice (i.e. T1, T2, T3, MZ, GC, B1a, B1b etc).

Given that GABA is produced at the highest levels by IgA+ plasma cells in the lamina propria of the gut, do mb1Cre-gad1fl/fl mice have changes in gut Tregs or gdT cells?

Author Rebuttals to Initial Comments:

Point-by- point responses

Referee #1

The authors begin with a clever basic experiment in which they performed a classic mouse foot-pad immunization and then brought to bear state-of-the-art technologies to compare the metabolites in the ipsilateral lymph node (LN) near to the immunization site versus those in the contralateral resting LN. They found 200 metabolites that differed significantly between these LNs, and a stand-out difference was activation of the glutamate pathway in the ipsilateral LN. Next, they assessed the contribution of different immune cell subsets to these metabolite differences by repeating the immunization studies with mice lacking T cells, B cells, or all mature B and T cells. Interestingly, the LN of B cell-deficient mice had the most distinct metabolic changes, and surprisingly, GABA, which heretofore was not known to be synthesized by B cells, was the major metabolite that was upregulated in the ipsilateral LN in a B cell-dependent manner. These results suggest that antigenic stimulation induces B cell synthesis of GABA. Using labeled glutamine, the authors traced its catabolism in activated B cells and found that glutamine was converted to glutamate (the substrate of the GABA synthetic enzyme glutamic acid decarboxylase (GAD65/67)) and that labeled GABA was almost exclusively detected in B cells, but not T cells. These data suggest that antigenic stimulation increases GAD enzymatic activity in B cells leading to greater production of the chemical transmitter GABA.

They then examined GABA's effects on anti-tumor responses using a common cancer model in which B cells are known to inhibit anti-tumor responses. As expected, B-cell deficient mice controlled tumor growth better than wild-type mice, and histological analysis showed expansion of tumor-infiltrating CD8+ T cells with enhanced cytotoxic and inflammatory markers. Implantation of a slow GABA-releasing pellet led to an increase in tumor size in B-cell deficient mice compared to controls, which was accompanied by reduced frequency of CD8+ T cells and their production of cytotoxic and inflammatory markers.

Importantly, they also observed that tumor-associated macrophages (TAMs) in B-cell deficient mice had enhanced expression of the TNF-signaling pathway compared to wild-type mice, and that this phenotype was dampened when B cell-deficient mice received GABA. The findings that the modulation of TNF pathways is B cell-dependent and related to changes in mitochondrial respiration are novel and interesting.

Finally, taking advantage of mice with conditional deficiencies in B cell GABA production (GAD67^{-/-} mice) they show that 1) their B cells have reduced intracellular GABA levels, 2) tumor cells implanted into these mice have significantly reduced growth, and 3) their tumor-infiltrating CD8+ T cells had enhanced cytotoxic and pro-inflammatory properties. Together, these results indicate that B-cell-produced GABA significantly contributes to limiting anti-tumor responses. The use of state-of-the-art technologies in this report is impressive and led to new observations that have previously gone unnoticed. Their major observation, that B-cell-produced GABA significantly contributes to limiting anti-tumor responses is a novel and unexpected finding. It points to new approaches to increase the efficacies of immunotherapies for solid tumors. According, the report will be of wide interest to the Nature readership at a basic science level as well as for its potential clinical applications.

Thank you very much for summary of our study, your appreciations and very positive comments.

Major concern:

The authors rule out a direct effect of GABA on CD8⁺ T cells based on a study shown in Extended data Fig. 6. There is, however, long-standing evidence that CD8⁺ T cell functions are directly modulated by activation of their GABAA-Rs. First, GABA was found to modulate CTL activity [1]. Second, electrophysiological studies by Birnir and colleagues [2] observed that physiological levels of GABA activated GABAA-R channels on CD8⁺ T cells at a single-cell level using patch-clamping. These GABA-Rs displayed channel conductance that was typical of extra-synaptic GABAA-Rs. GABA's effects were enhanced by a GABAA-R positive allosteric modulator further confirming the effects were GABAA-R mediated. These results, obtained by a leading GABA-R electrophysiology lab, provide strong functional evidence of a direct action of GABA on CD8⁺ T cells. Additionally, these authors found that low levels of GABA inhibited CD8⁺ T cell proliferation, similar to reports from many labs that GABA modulates the proliferation and cytokine/chemokine secretion of CD4⁺ T cells. Third, single-cell transcriptome data suggests that CD8⁺ T cells express GABA-Rs since their GABA-R subunit expression is essentially the same as that of CD4⁺ T cells (e.g., [3] and ImmGen data banks). Finally, GABAA-R agonist administration induces CD8⁺ Tregs *in vivo* [4, 5], although that effect could be indirect. Hence, GABA can act directly on CD8⁺ T cells and this notion should be carried throughout the manuscript when interpreting their observations.

The author's findings shown in extended data Fig 6 indicate that GABA treatment actually tended to reduce CD8⁺ T cell secretion of IFN γ and TNF *in vitro*. If the authors wish to provide contrary evidence to reports of GABA directly modulating CD8⁺ T cells, they should further support their negative observations by testing a dose range of GABA (100 nM-1 mM, n=6-7 samples/group, in duplicate or triplicate) in their cytokine and proliferation assays.

We share the reviewer's interest in GABA acting directly on CD8⁺ T cells as in the previous publications listed. From the very beginning of our study we tested this possibility and cultured purified CD8⁺T cells with GABA or a selective GABA_A receptor agonist, muscimol. While muscimol treatment clearly and consistently inhibited activation and proliferation of CD8⁺T cells, we did not observe similar effects with GABA, by reasons that are still obscure. Hence, the hesitation to add the data in the initial form of the manuscript. We now incorporate such data, including the GABA titration experiment suggested in the revised manuscript (**Extended Data Fig. 8e, f, g**).

To confirm that GABA_A receptors modulate anti-tumor immunity, we performed *in vivo* studies using a GABA_A receptors blocker. We found that the blockade of GABA_A receptors in WT mice by repeated injection of very low dose of picrotoxin, considerably limited the tumor growth and enhanced the cytotoxic activity of TIL CD8⁺ T cells (**Fig. 3d, e**). We also analyzed the transcriptome profiles of TAMs isolated from WT mice or WT mice injected with picrotoxin, and show enhanced inflammatory features of TAMs in the picrotoxin treated group (**Fig.3h, Extended Data Fig. 10e**).

We also show that *in vitro*, neither GABA nor picrotoxin treatment affected the proliferation and viability of MC38 cells (**Extended Data Fig. 7**).

However, addition of picrotoxin significantly enhanced the effect of anti-CD3/CD28-induced Ca²⁺ mobilization in mouse CD8⁺ T cells and human Jurkat T cells stimulated *in vitro*, clearly indicating GABA_A receptors on the surface of T cells modulating a pivotal signaling pathway (**Extended Data Fig. 8a, b**).

We thank this reviewer for the encouragement to pursue these experiments, confirming and building on previous publications, which were included in the reference list.

Moderate concerns:

A) It is standard practice when studying GABA-Rs to include controls with GABAA-R and GABAB-R-specific agonists and antagonists to verify the results are due to effects on GABA-Rs and not off-target effects, and to determine whether observations are GABAA-R or GABAB-R mediated. It would enhance the manuscript, but it is not essential, that at least one of their novel *in vitro* findings was validated using the GABAA-R agonist muscimol, the GABAB-R agonist baclofen, along with GABA combined with bicuculine or saclofin (GABAA-R and GABAB-R antagonists, respectively).

Following the suggestion, we analyzed the transcriptome profiles of GABA M-IL-10 macrophages differentiated *in vitro* in the presence of picrotoxin. GABA_A receptors were clearly involved in these transcriptomic changes, because the addition of picrotoxin partially reverted the effect of GABA on M-IL-10, including the IL-10 transcripts (**Extended Data Fig. 13b**).

Indeed, as this reviewer suggests we do not rule out additional GABA_B-receptor mediated effects, and have generated some preliminary data to this effect. We include this preliminary data as **Annex A**, which we request be considered confidential as future research by our laboratory aims to confirm and broaden these studies. We hope the non-essential nature of this particular suggestion from Reviewer 1 can be addressed by the informal access to these results without incorporation of the manuscript in its current scope.

B). The author's granular studies provide additional depths of knowledge on the actions of GABA on immune cells. For instance, they performed differential gene expression analysis that showed that GABA modulates macrophages toward anti-inflammatory responses. Please discuss these new findings in the context of past reports showing that GABA modulates APC toward anti-inflammatory responses. For instance, studies in the autoimmune and parasitology fields have shown that rodent and human APC express both GABAA-Rs and GABAB-Rs and both GABAA-R [4-10] and GABAB-R [3, 11-14] -specific agonists inhibit their inflammatory activities. Along this line, it is likely that APC in the tumor microenvironment secrete GABA, e.g., their reference 10 showed that macrophages and DC express GAD65 and secrete GABA. The authors should incorporate these findings into their interpretations and discussion.

Thank you for considerate remark and suggestion. We revised the text, added new data on GABA receptors on macrophages as mentioned above and also included previous publications mentioned.

“Activation of GABAergic signaling by GABA_A-receptor agonists has been shown to diminish the production of inflammatory cytokines by macrophages in an autoimmune disease model, confirming the presence of functional GABA receptors on macrophages¹⁹. We found that TAM isolated from WT mice after repeated injection of picrotoxin had upregulated transcripts related to Ca signaling and inflammatory cytokines, such as IFN- γ targeted genes (Fig.3 h, Extended Data Fig. 10e). Genes related to translation,

cell cycle, mitochondria and genes targeted by IL-10R were downregulated by picrotoxin (Fig.3 h, Extended Data Fig. 10e).”

C) GABA also inhibits Th1 and Th17 cells, while also promoting CD4+ and CD8+ Tregs ([15, 16] and references therein). Notably, administration of GABA-R agonists promotes the spreading of IL-10-secreting autoreactive T cell responses, while inhibiting the spreading of inflammatory T cell autoreactivities, to autoimmune target tissue antigens [15, 17]—these same actions are also likely to impede the spreading of anti-tumor effector responses among tumor antigen determinants. All of these effects on CD4+ T cells can directly attenuate CD4+ T cell-mediated tumor cell killing, as well as reduce help for CD8+ effector T cell responses, and should be made part of their interpretation and discussion.

We agree with the reviewer, and therefore included in the discussing the following sentence: “This research builds on a body of work describing GABA modifying the function of mature lymphocytes, such as CD4+ T cell effector subsets in the periphery, as well as developing cells, such as hematopoietic precursors in the bone marrow. For many of these studies, the source of GABA remained unclear^{14,32-36}.”

D) Please check the level of GABA in MC38 cells. Positive controls could include mouse brain cells and activated B cells, and negative controls could include HEK 293, HELA, or COS cells.

We performed these measurements, including GABA measurements in *ex vivo* sorted MC38 cells. As shown in **Figure 1 for Reviewer 1**, *in vitro* cultures, the GABA contents were comparable to the levels found in *ex vivo* IgA plasma cells in all tumor cell lines-including the proposed negative controls HEK 293 and HELA cells. However, *ex vivo* sorted MC38 tumor cells contained very little GABA.

Figure 1 for Reviewer 1

To further expand discussion along this line, it was also very puzzling how MC38 cells could make GABA *in vitro*, as we could not detect the transcripts for any of the enzymes converting glutamate (Gad1, Gad2) or putrescine (Aoc1) to GABA. The possible explanations are that these tumor lines avidly import GABA present in the media, or alternatively they could make GABA by ways yet unknown in mammalian cells. Although these issues of metabolome differences between *in vitro* and *in vivo* tumor cells are interesting, they are beyond the scope of the current manuscript.

Minor issues:

The text makes it seem that there are stark differences in GABA levels between B cells and T cells. However, their data (Extended data Fig. 3A) shows that GABA levels in T cells are only about 40% lower than that of B cells. If B cell GABA has such an immunological impact, it begs the question what is T cell GABA is used for? I understand that the authors observed that essentially no labeled glutamine ends up as GABA in activated T cells. However, GABA can be also made from putrescine ([3] and references therein) and can be taken up from the extracellular environment by GABA transporters. Their strong statements that T cells do not make or use GABA do not seem warranted by the information on hand.

We agree that T cells also contain GABA, and the various measures included in the manuscript show ratio of GABA expression with respect to B cells is dependent on the activation status of the cells used for metabolic analysis. However, based on the global metabolomic landscape of lymph nodes in immunodeficient mice and newly added imaging data, we think it is fair to say that GABA is produced more at a per cell level by B cells activation, but even more strikingly, GABA characterizes B cell zones of lymph nodes, making it an exciting first target for the functional experiments we present here. We certainly agree that there are many more questions to be solved in relation to the production, uptake or utilization of GABA in T cells, and also in many other cells and are greatly interested in pursuing these actions in future studies, but we hope this reviewer will agree these may be outside the scope of the current manuscript. In the current draft we have removed all strong statements as indicated by the reviewer.

Please inform this reviewer why they chose to knock out GAD67 and not GAD65 or both? Their reference 10 showed that mouse APC make GAD65 and GABA. But GAD67 is of interest since it releases GABA in non-vesicular fashion. In any case, they should bear in mind that cytoplasmic GABA can provide ATP via the GABA shunt especially in oxygen-poor conditions, and so GAD67-deficiency doesn't just affect GABA secretion.

The reason we chose GAD67 was that we could not detect transcript for GAD65 in B cells and T cells (we included the data in **Extended Data Fig. 4**).

We agree with this reviewer that GAD67 deficiency might impact on other aspects of B cell biology and we are planning to investigate such possibilities. However, so far, we can confidently say that the frequencies and numbers of B cells, myeloid cells and their precursors in the bone marrow of *mb1^{cre/+}-gad1^{fl/fl}* mice appear to be normal (**Extended Data Fig. 15**). Also, the frequencies and numbers of various B cell subsets were comparable to those observed in *mb1^{cre/+}-gad1^{fl/+}* mice. We were also somehow disappointed to find normal

IgA plasma cells compartment in *mb1^{cre/+}-gad1^{fl/fl}* mice (**Extended Data Fig. 16**). However, as the reviewer pointed out there are many other possibilities for future studies which we now list in the modified discussion text.

The authors make relative comparisons. For example, they say that GABA levels in GAD67^{-/-} B cells were reduced to the “minimal levels observed in T cells”. As noted above, the GABA in T cells was only about 40% less than B cells. Relative levels are not useful since GABAergic neurons or macrophages could also be used as reference points. Please discuss the levels of GABA in absolute units.

New measurements were made to quantify the absolute concentration of GABA in B and T cells isolated from lymph nodes of *mb1^{cre/+}-gad1^{fl/+}* and *mb1^{cre/+}-gad1^{fl/fl}* mice, and these are now presented in **Extended Data Fig. 17a**.

They note that GABA treatment had no effect on tumor size in wild-type mice and they speculate that B cell-derived GABA production saturates the system such that exogenous GABA does not impede anti-tumor responses. Please note that GABA has been found to inhibit, or conversely, promote the growth of a few different solid tumors, and some mechanistic insights have been obtained (reviewed in [18]). These findings suggest that the effect of GABA on tumors is dependent on the cancer cell phenotype and its intracellular signaling pathways, and not B cells. Along that line, I know of no clinical evidence of commonly prescribed GABA-R modulators modulating cancer incidence or progression.

We agree, and have altered this statement to reflect the possibility that the endogenous GABA production may be derived from various possible cellular sources, such as tumor cells. Additional data present in **Extended Data Fig. 7** now assesses the effect of a low-dose of the GABA_A receptor antagonist picrotoxin, confirming that GABA signaling can modulate tumor immunity in the MC38 model, while it does not appear to affect the growth of MC38 cells themselves.

Line 83 states GABA was the most upregulated metabolite in ipsilateral vs. contralateral. Yet, line 105 states GABA was found at elevated levels in B cells from the contralateral side. Please clarify.

We apologize for the confusion—we hope we clarified in the text.

“GABA was also detected in contralateral LN of WT and *cd3e^{-/-}* mice, albeit at lower levels compared to ipsilateral LNs (Fig. 1e). However, very little GABA could be detected in either ipsilateral or contralateral LN from B cell-deficient mice (*muMt^{-/-}*) and *rag1^{-/-}* mice, indicating that high GABA levels in LN tissue represents a B cell signature (Fig. 1e).”

Referee #2

Zhang et al. B cell derived GABA elicits anti-inflammatory macrophages and limits anti-tumor cytotoxic responses

This paper starts out with an unbiased look at what metabolites are made in reactive versus non-reactive mouse lymph nodes. Among a variety of metabolites, GABA was prominently made primarily by B cells and less by T cells or myeloid cells. B cell activation seemed critical for GABA expression by B cells. In the MC38 tumor model, tumor growth was reduced in uMT mice and was partially restored by adding exogenous GABA. CD8 T cell function (GZB, PRF, IFN γ and TNF) was increased in uMT mice, and exogenous GABA returned T cell activation back to WT levels. Tumor associated macrophages had very different transcriptomes in WT and uMT mice, but it was less clear whether exogenous GABA restored the B cell phenotype. Nevertheless, genes like TNFRSF1b, IFN γ R, IL-1b were clearly regulated in a GABA-dependent fashion. BM-derived macrophages cultured with GABA had higher folate R expression, increased OXPHOS, less inflammatory cytokines and higher IL-10 and reduced MHC expression. IL-10 was a prominent factor in reducing T cell activation by macrophages. Finally, mice lacking GAD1 specifically in B cells exhibited delayed tumor growth and enhanced T cell function.

Thank you very much for the positive comments and summary of our findings.

The presence or absence of B cells has a much profound effect on tumor growth and macrophage function than can be accounted for by GABA (Fig 3A and D). In particular, the gene expression profiles of TAMs from uMT do not look like they were restored to WT with the addition of GABA (Fig 3D). Nevertheless, the TNF target genes and the OXPHOS genes are at least partly regulated by GABA. Is this a very specific convergence of GABA signaling on TNF signaling pathway?

We agree that the absence of B cells had a much more profound effect than the specific inactivation of Gad1 in B cells. Regarding the convergence of GABA signaling on TNF signaling, this is an extremely interesting question also pointed out by the Referee 1, and which we have addressed experimentally.

We isolated monocytes from tumor-bearing WT mice and differentiated them *in vitro* in the presence of GABA. The cells were then stimulated with TNF- α or for comparison with IL-1 β , prior to evaluation of NF- κ B activation. As shown in **Fig.4e**, GABA greatly reduced the nuclear localization of total p65 induced by TNF- α , while only partially attenuated its translocation induced by IL-1 β .

In Fig 3B, the data is expressed as % of T cells in the CD45+ population. Given that the B cells are missing, the CD45 population is not the appropriate denominator in the uMT mice. This data needs to be expressed as actual number of T cells / tumor or number of CD8 T cells/ tumor.

We thank Reviewer 2 for this suggestion and now present the total cell numbers.

In Fig 3C, the FACS plots need to contain the % of cells + or – SD in each quadrant or gate. The graphs do not make sense – is this a % or is the data normalized to control. It looks like the data is normalized to 1 in WT. This should be represented as % of the CD8+ T cell

population

With thanks to the reviewer, we have changed the plots to indicate the percentage as a proportion of CD8⁺ T cells as suggested. We apologize for the confusion.

Fig 4H needs a statistical analysis – AUC?

We thank the reviewer for this suggestion, this analysis has been added.

Fig 4i The gated areas need to be labeled with percentage + and – SD. Stats?

We changed the plots accordingly.

Is the real functional impact of B cell-derived myeloid cells in the lymph node or in the tumor? The data is all about the tumor, but it may be that the T cells and the myeloid cells are messed up in the lymph node and never make it to the tumor. Maybe the T cells are excluded from the tumor.

This is an extremely interesting point. We performed FACS analyses and compared the numbers and functional properties of CD4⁺ and CD8⁺ T cells isolated from the tumor draining lymph nodes of WT mice, *muMt*^{-/-} or *muMt*^{-/-} mice implanted with GABA pellet. We did not observe any obvious differences in T cells between these groups, we include this data for Reviewer's 2 interest (**Figure 1 for Reviewer 2**) and would be happy to include this data in the manuscript if it is considered informative. However, we suggest that more rigorous and dynamic measurements may be required to confidently conclude whether T cells migration into the tumor is affected by B cells, plasma cells or macrophages in the lymph nodes, but hope this may be addressed in more detail in future studies.

Figure 1 for Reviewer 2

New data from Shao 2021 Blood PMID: 32881992 show that GABA acts on B cell development and that GABA agonists enhance hematopoiesis. Are myeloid macrophages and

B cells normal in GABA-supplemented mice?

Thank you for pointing out this interesting paper, indicating Gad1 but not Gad2 expression in the bone marrow particularly enriched in B cells and that GABA agonists enhance the proliferation of hematopoietic/progenitor stem cells.

We found comparable numbers of hematopoietic/progenitor stem cells (LSK, HSC, CMP or GMP) in WT mice supplemented with GABA (**Figure 2 for Reviewer 2**) and in mice with B cell specific deletion of Gad1 (**Extended Data Fig. 15a**).

To develop this line further, we think that the initial conditioning of monocytes most likely takes place in the bone marrow, where we found that B cells, plasma cells and monocytes closely co-localized. We discuss these possibilities in the modified text of the discussion. We have presented the first evidence that B cells are a dominant source of GABA in the peripheral tissues studied, and also revealed that GABA impacts on fundamental processes of macrophage physiology and facilitates macrophages polarization toward an anti-inflammatory phenotype.

Figure 2 for Reviewer 2

Referee #3

This is a very interesting study that reveals anti-inflammatory and pro-tumor effects of B cell derived GABA in mice. GABA is produced at high levels by plasma cells in the lamina propria of the gut, and to a lesser extent by all B cell subsets. GABA counteracts the anti-tumor effect of B cell-deficiency, at least in part by indirectly promoting formation of cytotoxic T cells. Using bone marrow-derived macrophages, the authors show that GABA leads to the formation of anti-inflammatory IL-10-secreting macrophages, that can dampen CD8 T cell cytotoxicity in an IL-10-dependent manner. In mice lacking Gad1 in B cells (that cannot form GABA), tumor formation is greatly reduced. The results are exciting and open up new avenues of research. The conclusions could be strengthened by addressing the issues listed below.

Thank you very much for the encouraging remarks and summary of our findings.

Issues to address.

While most publicly available databases reveal Gad1 RNA expression in mouse B cells, this is not apparent in human B cells. Have the authors looked for GABA production by human B cells? If this pathway is not present in humans, this should be mentioned.

We appreciate raising this issue. Indeed, we have found that not only mouse but also human B cells express GAD1 transcripts. For both mouse and human B and T cells, no transcripts for GAD 65, or Gad2/GAD2, could be detected. We now show the transcripts for glutamate decarboxylase (GAD) 67 or Gad1/GAD1 in mouse and human B cells and T cells (**Extended Data Fig. 4**).

Giving the strong interest from this reviewer and also from the editor, we further performed experiments testing the capacity of human B cells to convert glutamine into GABA, performing C13 glutamine tracing experiments with human tonsil and blood-derived B cells, stimulated in vitro in two different conditions. The results clearly indicate that stimulation of human B cells facilitated the conversion of glutamine to GABA (**Fig. 2e**) and increased the levels of both intracellular and secreted GABA derived from labeled glutamine (**Fig. 2f**, **Extended Data Fig. 5c**). We present new data showing GABA overlapping with B cell follicles in human tonsil tissue (**Extended Data Fig. 5d**) or infiltrating B cells and IgA plasma cells in human renal cell cancer tissue (**Extended Data Fig. 18**).

We are very grateful for this suggestion, as the inclusion of these experiments emphasizes the relevance these findings may have for humans.

The authors draw the conclusions on direct effect on macrophages from cultures in which total bone marrow cells are incubated with GABA and M-CSF from day 0. These experiments do not exclude the possibility that GABA is acting on other bone marrow cells to influence macrophage differentiation. To conclude GABA acts directly on macrophages, purified primary macrophages obtained from spleen or peritoneal cavity could be used.

Although the cultures we performed are considered standard in the field, we agree with the reviewer that there could be other cells that could potentially interfere the macrophage

differentiation. We confirmed the validity of our presented data with purified monocytes, showing that GABA-IL-10 conditioned IL-10 producing macrophages (**Extended Data Fig.12b**).

Regarding the impact on mature macrophages differentiated *in vitro* or isolated from peritoneal cavity (PC), the IL-10 transcripts were not changed by the addition of GABA, although we did observe an increase in arginase-1 in matured BMDM (ma-BMDM). We present these data as **Figure 1 for Reviewer 3**.

To even further address this issue, we performed conditioning experiments with sorted blood monocytes from humans. We now added data showing that GABA increased the cell number and the surface expression of folate receptor β (FR β) in human macrophages differentiated from purified blood monocytes under neutral conditions (M-0), similar to mouse macrophages (**Fig. 4a, right graph, b right panels, Extended Data Fig. 11b, right graph**). These results we believe greatly enhance the strength of our conclusions and relevance of the GABA for humans,

Figure 1 for Reviewer 3

In line with the above, to show that the changes in tumor infiltrating CTLs are a consequence of GABA acting on macrophages rather than on the tumor itself, or on other haemopoietic or endothelial cells, tumor growth could be evaluated in *mb1.Cre-gad1.fl/fl* mice after macrophage depletion.

We agree that we cannot exclude that GABA could act directly on tumor cells, and this was also mentioned by the Reviewer 1 in the minor comments section. We could not perform the depletion experiments with the *mb1^{cre/+}-gad1^{fl/fl}* mice because of a limited number of mice. However, we performed two type of experiments addressing the involvement of macrophages in anti-tumor responses modulated by GABA.

In the first set of experiments, we depleted macrophages in WT and *muMt^{-/-}* mice and found opposite trajectory for tumor growth. While in WT mice macrophage depletion reduced in tumor growth, bigger tumors were observed in *muMt^{-/-}* mice undergoing similar treatment. We present this data pointing to distinct immune-regulatory properties of macrophages in the presence or absence of B cells as **Extended Data Fig. 9**.

In the second set of experiments, we co-injected macrophages differentiated *in vitro* together with MC38 into WT mice. The co-injection of GABA M-IL-10 with MC38 facilitated the tumor growth compared to control mice injected with MC38 alone or MC38 + M-IL-10, further supporting the anti-inflammatory nature of macrophages differentiated in the presence of GABA (**Extended Data Fig. 12g**).

We thank this Reviewer for bringing up this issue. We believe we obtained important data that solidify the involvement of macrophages and how GABA might modulate the macrophage properties and anti-tumor responses.

Have the authors evaluated the effects of GABA on CD4⁺ Tregs? Dampening their numbers or function in tumors of MuMT mice may be an alternative or additional explanation for promoting tumor growth.

We evaluated the number and frequencies of Foxp3⁺ T cells, and also measured the Foxp3 expression levels of Foxp3⁺ as indicator of their suppressive function. We present these data as a **Figure 2 for Reviewer 3**.

There were no significant differences of Tregs in tumor sites in WT or *muMt*^{-/-} mice plus or minus GABA pellets. As expected, the frequencies of Foxp3⁺ T cells were higher in draining LNs of *muMt*^{-/-} mice, but GABA supplementation did not have a significant effect, although there was a tendency towards higher Foxp3 expression following GABA that did not reach a threshold for significance. The direct effect of GABA on suppressive capacity of Treg would need to be further evaluated, but we hope this reviewer will agree this may be outside the scope of the current manuscript.

Figure 2 for Reviewer 3

Have the authors looked for expression of PD-L1 on tumor cells (not just macrophages) before and after GABA injection in the MuMT mice to exclude direct actions on the tumor itself?

We performed *in vitro* experiments showing that neither GABA nor picotoxin treatment affected the proliferation, viability (**Extended Data Fig. 7**), or PD-1, PD-L1 or PD-L2 expression of MC38 cells (**Figure 3 for Reviewer 3 (upper)**). We also have not observed an obvious difference of PD-L1 expression on tumor cells after picotoxin injection in WT mice or GABA pellet implantation in the *muMt*^{-/-} mice (**Figure 3 for Reviewer 3 (lower)**).

Figure 3 for Reviewer 3

Only the bone marrow B cell compartment has been characterized in depth in Mb1-Cre.Gad1 fl/fl mice. It would be important to show normal peripheral B cell compartments in these mice (i.e. T1, T2, T3, MZ, GC, B1a, B1b etc). Given that GABA is produced at the highest levels by IgA⁺ plasma cells in the lamina propria of the gut, do mb1Cre-gad1 fl/fl mice have changes in gut Tregs or gdT cells?

We presented the phenotypic characterization of various B cells and T cells subsets in the spleen, lymph nodes, Peyer's patches, lamina propria, intraepithelial lymphocytes and the peritoneal cavity of *mb1^{cre/+}-gad1^{fl/+}* and *mb1^{cre/+}-gad1^{fl/fl}* mice, confirming the similarity in B cell numbers observed in bone marrow (**Extended Data Fig. 16**).

Redacted

Reviewer Reports on the First Revision:

Referee #1 (Remarks to the Author):

I thank the authors for their efforts to respond to the reviewers and for the many additional studies they performed which were very responsive to the review.

The additions that the authors have made to the revised manuscript are impressive, including 1) visualization of GABA in lymph nodes using imaging mass spectroscopy, 2) studies of human B cells showing that they synthesize and secrete GABA (similar to the findings in mice). 3) antagonism of GABAA-Rs limited tumor growth and enhanced the activity of tumor-infiltrating CD8+ T cells 4) CD3/CD28-induced Ca²⁺ mobilization in murine and human CD8+ T cells by GABAA-R antagonist treatment, and 5) GABAA-R antagonist enhanced expression of inflammatory genes in tumor-associated macrophages. Together with the previous findings, the studies are quite a tour-de-force.

In regards to their new studies:

1. The author's new studies with muscimol and picrotoxin clearly point to the role of GABAA-Rs in their observations. Oddly, they observe little effect of GABA at 0.1, 10, and 1000 μ M. This contrasts with published studies on GABAs' effects on CD8+ T cell electrophysiology and proliferative responses. The 100-fold steps in the GABA levels they tested may have missed effects if the dose-response was not linear. If the authors have not tested GABA dosages between the range of 200-800 μ M I encourage them to do so.

2. The revision contains new findings that patients with rheumatoid arthritis have elevated plasma levels of GABA that were positively correlated with disease activity scores and autoantibody titers. This contrasts with an earlier study that found significantly decreased levels of GABA in rheumatoid arthritis vs. control subjects (PMID: 32130577). Similar reductions in GABA levels were reported in osteoarthritis patients (PMID 29631251) in which B cells play a role (PMID: 23360836). Please make the readers aware of the contrasting published findings in RA patients.

Finally, on a minor note, since readers are more familiar with the terms "GAD65" and "GAD67" and GAD67^{-/-} and GAD65^{-/-} from past studies, I suggest favoring these notations when possible over their gene names.

Reviewed by Daniel Kaufman

Referee #2 (Remarks to the Author):

The authors have responded appropriately to my prior comments

Referee #3 (Remarks to the Author):

The authors have addressed all of the questions and concerns and the new data has strengthened their conclusions.

Author Rebuttals to First Revision:

Referee #1

I thank the authors for their efforts to respond to the reviewers and for the many additional studies they performed which were very responsive to the review.

The additions that the authors have made to the revised manuscript are impressive, including 1) visualization of GABA in lymph nodes using imaging mass spectroscopy, 2) studies of human B cells showing that they synthesize and secrete GABA (similar to the findings in mice). 3) antagonism of GABAA-Rs limited tumor growth and enhanced the activity of tumor-infiltrating CD8+ T cells 4) CD3/CD28-induced Ca²⁺ mobilization in murine and human CD8+ T cells by GABAA-R antagonist treatment, and 5) GABAA-R antagonist enhanced expression of inflammatory genes in tumor-associated macrophages. Together with the previous findings, the studies are quite a tour-de-force.

In regards to their new studies:

1. The author's new studies with muscimol and picrotoxin clearly point to the role of GABAA-Rs in their observations. Oddly, they observe little effect of GABA at 0.1, 10, and 1000 uM. This contrasts with published studies on GABAs' effects on CD8+ T cell electrophysiology and proliferative responses. The 100-fold steps in the GABA levels they tested may have missed effects if the dose-response was not linear. If the authors have not tested GABA dosages between the range of 200-800 uM I encourage them to do so.

We thank this Reviewer for the kind praise and very insightful suggestions. We performed the requested experiment and can confirm the Reviewer was correct. We did observe a clear reduction of inflammatory cytokine production by CD8+ T cells in the presence of the indicated GABA concentrations. We are grateful to the Reviewer for this helpful suggestion, as the result has greatly increased our understanding of the mechanisms underlying GABAs impact on tumor immunity in our model, and resolved outstanding questions for us about how our findings compared to previous studies in the field. We added the results in Extended data Fig.8c and removed the previous FACS proliferation results.

2. The revision contains new findings that patients with rheumatoid arthritis have elevated plasma levels of GABA that were positively correlated with disease activity scores and autoantibody titers. This contrasts with an earlier study that found significantly decreased levels of GABA in rheumatoid arthritis vs. control subjects (PMID: 32130577). Similar reductions in GABA levels were reported in osteoarthritis patients (PMID 29631251) in which B cells play a role (PMID: 23360836). Please make the readers aware of the contrasting published findings in RA patients.

Thank you the Reviewer for their advice, as the resulting increased statistical rigour applied to this experiment has increased our confidence in our findings. We now show a non-parametric Spearman test for the correlation results in Fig.1f ($p=0.07$, $r=0.18$). We are confident from other sets of data that activation of B cells correlates with GABA elevation in plasma, but we let the Editor and Reviewer the decision if the data presented in Fig1f should be included or not.

We note that the cited papers did not show the correlation with disease score or autoantibody levels, and so this finding does not contradict previous studies *per se*. We have added a line to this effect in the revised manuscript.

“In contrast to previous studies^{4,5}, we found a positive correlation between plasma GABA levels with the disease activity scores and autoantibody titers in patients with rheumatoid arthritis, suggesting that GABA is indicative of B cell activation in humans (Fig.1f).

Finally, on a minor note, since readers are more familiar with the terms “GAD65” and “GAD67” and GAD67-/- and GAD65-/- from past studies, I suggest favoring these notations when possible over their gene names.

We agree that the terms are complicated. We now make sure in the subheading that we inactivated GAD67 specifically in B cells. However, we have retained our nomenclature for the cre/lox mouse breeding strains, according to the standard usage of gene names to describe these mice in the field. If the Nature editorial staff has a preference for how this terminology should be handled, we would be happy to make additional changes.

Referee #2

The authors have responded appropriately to my prior comments

Thank you very much for your comments and appreciation.

Referee #3

The authors have addressed all of the questions and concerns and the new data has strengthened their conclusions.

We very much appreciate the time, effort and suggestions and reading once more the revised manuscript.